# Optical properties of size and chemical fractions of suspended particulate matter in littoral waters of Quebec

Gholamreza Mohammadpour[1], Jean-Pierre Gagné[1], Pierre Larouche[2], Martin A. Montes-Hugo[1*]

[1]Institut des Sciences de la Mer de Rimouski, 310 Alleé des Ursulines, Office P-216, Rimouski, Québec, Canada, G5L 3A1

[2]Institut Maurice-Lamontagne, Pêches et Océans Canada, Mont-Joli, Québec, Canada, G5H 3Z4

*Correspondence to*: Martin A. Montes-Hugo (martinalejandro_montes@uqar.ca)

**Abstract.** Empirical mass-specific absorption ($a_{SPM}^*$) and scattering ($b_{SPM}^*$) coefficients of suspended particulate matter (SPM) were measured for four size fractions (i = 0.2-0.4 µm, 0.4-0.7 µm, 0.7-10 µm, and >10 µm) in surface waters (i.e., 0-5 m depth) of the Saint Lawrence Estuary and Saguenay Fjords (SLE-SF) and during June of 2013. True absorption ($\sigma_a$) and

scattering ($\sigma_b$) cross sections for total particulate inorganic (PIM) and organic (POM) matter were also measured. Lastly, the response of two optical proxies (the spectral slope of particulate beam attenuation coefficient and mass-specific particulate absorption coefficient, hereafter $\gamma$ and *Svis*, respectively) to changes on particle size and chemical composition was examined. For the spectral range 400-700 nm, relatively low $a_{SPM}^*$ values (i.e., 0.01-0.02 m$^2$ g$^{-1}$) indicate large-sized particle assemblages with relatively high particulate organic carbon and chlorophyll a per unit of mass. Conversely, largest $a_{SPM}^*$

values (i.e., > 0.5 m$^2$ g$^{-1}$) corresponded with locations having relatively small-sized or mineral-rich particulates. Particle-associated iron likely explained the relatively high $a_{SPM}^*$(440) values in low-salinity environments of SF. The differential Junge slope of particle size distribution had a larger correlation with $b_i^*$ (Spearman rank correlation coefficient $\rho_s$ up to 0.37) with respect to $a_i^*$ ($\rho_s$ up to 0.32). Conversely, the ratio between PIM and SPM concentration had a stronger influence on $a_i^*$ ($\rho_s$ up to 0.50). Size spectrum (chemical composition) of SPM appears to be more important affecting relatively large (small)

particulates. The magnitude of $\gamma$ was sensitive to changes on size fractions of SPM mass. In LE locations, the magnitude of *Svis* was directly correlated with the mineral content of SPM. This may indicate a potential association between iron and inorganic enrichment of particles in areas of the estuary with a larger marine influence.

## 1 Introduction

The distribution of suspended particulate matter (SPM) in coastal and estuarine environments has a major influence on

several biogeochemical processes (e.g., phytoplankton blooms) (Guinder et al., 2009), ecosystem structure (e.g., food webs) (Dalu et al., 2016) and dispersion of pollutants (e.g., copper, mercury, polycyclic aromatic hydrocarbons) (Ma et al., 2002;

Ramalhosa et al., 2005). Light absorption by suspended particulates is essential for several photochemical processes related to the carbon cycle (e.g, photosynthesis, production of dissolved inorganic and organic carbon) (Estapa et al., 2012). Lastly, the concentration of SPM (CSPM) (Table 1) is an important variable for modeling thermodynamic processes and computing heat budgets (Löptien and Meier, 2011) due to its influence on underwater light attenuation (Morel and Antoine, 1994; Devlin et al., 2008).

Remote sensing allows synoptic mapping of SPM in littoral environments where the spatial and temporal variability of suspended particulates is relatively high. Indeed, optical measurements derived from spaceborne ocean color sensors are commonly applied for estimating CSPM based on visible (i.e., wavelength, $\lambda$ = 400-700 nm) (Miller and McKnee, 2004; Montes-Hugo and Mohammadpour, 2012) and NIR-SWIR (near-and short-wave infrared) ($\lambda$ = 700-3,000 nm) (Doxaran et al., 2002) spectral bands. Despite this progress, there is still a lack of understanding regarding how SPM microphysical characteristics (e.g., particle chemical composition and size distribution) relate to mass-specific inherent optical coefficients. This knowledge is essential for deriving more accurate remote sensing algorithms for estimating CSPM and developing new optical inversions for retrieving second-order attributes of SPM (i.e., chemical composition, size distribution). The optical characterization of particle size distribution (PSD) and/or composition in coastal and oceanic waters has been attempted based on four main methodologies: (1) analysis of spectral changes of inherent optical properties(Boss et al., 2001; Loisel et al., 2006), (2) empirical relationships between mass-specific optical cross sections and biogeo-physical characteristics of particulate inorganic matter (PIM) (e.g., mean diameter) (Bowers et al., 2009) and SPM (e.g. apparent density of particulates) (Neukermans et al., 2012), (3) optical inversions of different volume scattering functions (Zhang et al., 2014), and (4) changes on water leaving polarized reflectance (Loisel et al., 2008). A widely used methodology for estimating particle size spectra changes is the use of the spectral slope of particulate beam attenuation coefficient ($\gamma$) due to its relationship with the differential Junge slope of particle size distribution ($\xi$) (Boss et al., 2001).

Lastly, the biogeo-optical modeling of size and chemical fractions of SPM has a major scientific interest for understanding the dynamics of different mineral iron forms in coastal waters (Estapa et al., 2012) as particle-associated iron has two specific light absorption bands (wavelength, $\lambda$ = 360-390 nm and $\lambda$ = 400-450 nm). Also, Estapa et al. (2012) demonstrated that optical proxies such as the spectral slope of particulate absorption (*Svis*) within the visible spectral range ($\lambda$ = 400-700 nm) could be used for estimating dithionite-extractable iron and organic carbon content in marine samples. Iron can be part of organic (e.g., complexed forms) or inorganic (e.g., silicate sheets) particulates having a broad size range (e.g., from clays to amorphous aggregates) (Bettiol et al., 2008). Thus, the analysis of different fractions of SPM is essential for understanding the complex fate of iron in aquatic systems. Linking iron distributions with optical properties of size and chemical fractions of SPM may allow the development of proxies for mapping iron based on optical (*in water* and remote sensing) measurements. This is particularly advantageous for long-term monitoring projects as direct iron measurements are very expensive, difficult, and demand highly trained technicians.

The Saint Lawrence Estuary (SLE) and the Saguenay Fjords (SF) constitute a large sub-Arctic system characterized by relatively high concentrations of chromophoric dissolved organic matter (CDOM) (Nieke et al., 1997). Accurate remote sensing measurements of CSPM and SPM microphysical characteristics in these waters is crucial for understanding regional climate effects on coastal erosion (Bernatchez and Dubois, 2004) and occurrence of harmful algae blooms (Fauchot et al. 2008). However, in order to accomplish this task it is essential to know how mass-specific optical coefficients of suspended particulates are influenced by particle composition and size distribution changes. To our knowledge, mass-specific absorption and scattering coefficients of SPM size fractions have never been reported in the literature even though it has a practical application in biogeo-optical inversions and biogeochemical studies regarding the dynamics of trace metals.

This study has two main objectives: (1) to characterize the mass-specific absorption ($a_{SPM}$*) and scattering ($b_{SPM}$*) coefficients for four size fractions of SPM (0.2-0.4 µm, 0.4-0.7 µm, 0.7-10 µm, and >10 µm) and absorption ($\sigma_a$) and scattering ($\sigma_b$) cross sections for total particulate inorganic (PIM) and organic (POM) matter in different locations of the SLE-SF and during spring conditions, and (2) to establish relationships between mass-independent optical coefficients calculated in (1) and 'bulk' microphysical properties of particulates related to PSD and mineral content, and (3) to examine the response of two optical proxies ($\gamma$ and *Svis*) to changes on PSD and chemical composition.

This study is organized in three sections. In the first section, $a_{SPM}$*, $b_{SPM}$*, $\sigma_a$, and $\sigma_b$ coefficients are calculated for different optical environments of the SLE-SF that are characterized by a variable CDOM contribution to light attenuation and distinct particle assemblages. In the second section, the response of mass-specific optical coefficients and optical cross sections of SPM fractions to variations in PSD and mineral-content of suspended particulates is investigated. Lastly in the third section, the influence of PSD and mineral enrichment of particulates on $\gamma$ and *Svis* is examined. Also, spatial distributions *Svis* are interpreted in terms of salinity changes and potential particulate iron-rich environments.

## 2 Data and methods

### 2.1 Study area

The SLE can be divided in two main regions having contrasting biological productivity and bathymetry: the upper (UE) and the lower (LE) estuary (Levasseur et al., 1984). Non-algal particulates (NAP) and CDOM dominate the underwater light attenuation of UE waters (Nieke et al., 1997). This is in part related to the inflow of CDOM-rich and NAP-rich waters coming from the St. Lawrence River and Saguenay Fjord (Tremblay and Gagné, 2007; Xie et al., 2012). Unlike NAP and CDOM, contribution of phytoplankton to inherent optical properties increases towards the mouth of the SLE (Montes-Hugo and Mohammadpour, 2012; Xie et al., 2012).

The study of optical properties in SLE waters began during the late 80's. Babin et al. (1993) investigated the horizontal variability of the specific absorption coefficient of phytoplankton (i.e., absorption coefficient normalized by concentration of chlorophyll + phaeopigments) in surface waters during summer of 1989 and 1990. During the summer of 1990, Nieke et al.

(1997) studied the spatial variability of CDOM in terms of fluorescence and absorption spectra. Also, this study reported for the first time relatively high (up to 3 m$^{-1}$) particulate beam attenuation coefficients ($c_{SPM}$) and inverse relationships between salinity, $c_{SPM}$, and CDOM absorption coefficients ($a_{CDOM}$). Larouche and Boyer-Villemaire (2010) proposed remote sensing models for estimating PIM in SLE and Gulf of Saint Lawrence regions. Xie et al. (2012) showed inverse relationships between salinity and absorption coefficients of non-algal particulates and highlighted the extremely high $a_{CDOM}$ values (i.e., up to 5.8 m$^{-1}$ at $\lambda$ = 412 nm) along the Saguenay Fjord.

Historical studies performed during summer of 1975 suggest that size distribution of SPM differs between UE, LE and SF regions (Poulet et al., 1986). Based on surface samples, Poulet et al. (1986) found a dominance of relatively 'small-sized' (i.e., mode diameter < 10 µm) and 'large-sized' (i.e., > 30 µm) particulates over the UE and the mouth of the SLE, respectively. Conversely, the remaining locations of the LE were characterized by particulates having an intermediate size (i.e., 8-40 µm). In surface waters of SF and during spring months, SPM is commonly composed by very small particles (i.e., 2-3 µm) (Chanut and Poulet, 1979). However, this pattern is reversed during autumn. Several investigations point out that suspended particulates in SLE-SF regions are principally composed by inorganic matter (D'Anglejan and Smith, 1973; Larouche and Boyer-Villemaire, 2010). This mineral contribution varies between 60 and 95% of dry weight depending on the geographic location and period of the year (Yeats, 1988; Larouche and Boyer-Villemaire, 2010). Despite their important contribution, none of these studies reported mass-normalized optical coefficients for different size or chemical fractions of SPM nor an assessment of particle composition and size distribution effects on $a_{SPM}*$, $b_{SPM}*$, and optical cross sections of PIM and POM.

### 2.2 Field surveys

Discrete water samples for biogeochemical and optical measurements were obtained in 22 locations distributed throughout the SLE (N =17) and SF (N = 5) regions (Fig. 1). One discrete sample was obtained in each sampling locations but in site 6 where 2 measurements were made during June 3 and 6 of 2013. Samples corresponding to a sampling depth of 0-2 m were collected during June 3-9 of 2013 by using an oceanographic rosette equipped with Niskin bottles (volume = 12 L). For each sampling location, mass of different size fractions of SPM, optical coefficients for different SPM size fractions, and particle size distribution spectra were measured inside the wet lab of the vessel.

### 2.3 Biogeochemical analysis

The concentration of SPM (CSPM) in g m$^{-3}$ was measured gravimetrically after filtering a volume of seawater through pre-weighed GF/F filters (47 mm, average pore size = 0.7 µm, Whatman). The precision of CSPM determinations was 15% (Mohammadpour et al., 2015). The precision of 15% was computed as the percentage of ± 1 standard deviation with respect to the arithmetic average of weight corresponding to 10 replicas. Size fractionation of SPM into four size classes (>10 µm, 0.7-10 µm, 0.4-0.7 µm, and 0.2-0.4 µm) was done after sequentially filtering the original samples through pre-weighted membranes having a diameter of 47 mm and a pore size of 10 µm (Whatman, polycarbonate), 0.7 µm (GF/F, Whatman,

glass fiber), 0.4 µm (Whatman, polycarbonate), and 0.2 µm (Nucleopore, polycarbonate), respectively. The contribution of size fraction i to the total mass of SPM ($F_{SPM}^{i}$) was computed by normalizing their weight by the sum of weights corresponding to the 4 size fractions i. The mass of PIM was obtained after removing the organic fraction (i.e., POM) from the total mass of SPM as computed for CSPM determinations. The mass of POM was eliminated by combustion of GF/F

filters at 450°C and during 6 h. The concentration of POM was calculated as the difference between the dry mass of SPM and the dry mass of PIM. Based on Barillé-Boyer et al. (2003) factors and clay composition data obtained in the Saint Lawrence Estuary (D'Anglejan and Smith, 1973), the estimated error of PIM determinations due to dehydration of clays was 3.1%. Thus, PIM mass determinations has a maximum uncertainty of 18.1%. Notice that error in POM mass estimates is slightly greater than that associated to PIM mass (18.22% of loss on ignition PIM mass). The contribution PIM and POM to

SPM mass is $F_{SPM}^{j}$ where j superscript symbolizes PIM or POM, respectively.

**2.4 Optical measurements**

Total absorption (*a*) and beam attenuation (*c*) coefficient measurements were done on unfiltered and size-fractioned filtered water samples previously described in section 2.3. Discrete samples for optical coefficients were measured onboard by using an absorption-beam attenuation meter (ac-s, WetLabs, $\lambda = 400.3\text{-}747.5$ nm, average spectral resolution = 4 nm, path-length =

10 cm, accuracy $\pm 0.001$ m$^{-1}$). In order to minimize the presence of bubbles, a pump (ISMATEC MCP-Z) was used to gently circulate the samples through the ac-s tubes. Spikes on raw signal associated to bubbles were removed by visual inspection.. Residual scattering on absorption measurements was removed by applying a flat baseline at a reference wavelength of 715 nm (Bricaud and Stramski, 1990). This is a first order correction for scattering effects on non-water absorption coefficient estimates. Thus, the calculation of particulate absorption coefficients in this study is expected to have a bias with respect to

true values measured using absorption-meter instruments that are less influenced by particulate scattering (e.g., point-source integrating-cavity absorption meters) (Röttgers et al., 2013). Lastly, values of *a* and *c* were corrected by water temperature and salinity variations (Pegau et al. 1997). Spectral values of $a_{SPM}$ were derived by subtracting $a_{CDOM}$ and the absorption coefficient for seawater ($a_w$) to *a* at each wavelength. The contributions $a_{CDOM} + a_w$ were measured by using the *a*-tube (i.e., reflective tube) of the ac-s and after pre-filtration of total samples through a membrane having a pore size of 0.2 µm

(nucleopore, Whatman). Similar to $a_{SPM}$ calculations, the magnitude $c_{SPM}$ was computed after subtracting CDOM and seawater contributions to *c* as derived by using the *c*-tube (i.e., opaque tube) of the ac-s instrument. Lastly, particulate scattering coefficients ($b_{SPM}$) were derived by subtracting $a_{SPM}$ to $c_{SPM}$ values.
The particle size spectra within the size range 3-170 µm were measured on 'bulk' (i.e., without size fractionation) samples and by using a red laser (wavelength = 670 nm) diffractometer (LISST-100X, type B, Sequoia Scientifics) (Agrawal et al.

1991). LISST bench determinations were discrete and performed on board of the ship. Lab measurements were performed by using a chamber and a magnetic stir bar in order to homogenize the samples and avoid sinking of particulates. The optical path was covered with a black cloth to minimize ambient light contamination during the scattering measurements. The LISST-100X instrument can measure 32 scattering angles within an angular range of 0.08-13.5°, thus, particulates with a

diameter between 1.25 and 250 μm can be quantified. However only the interval 3-170 μm was analyzed due to variability of particle shape and refractive index in the first bins (i.e., < 3.2 μm) (Agrawal et al., 2008; Andrews et al., 2010), stray light effects in the first bins (Reynolds et al. 2010), and bias related to particle sinking in the last bin (i.e., 170-250 μm) (Reynolds et al. 2010). Measurements were made during a period of 3 minutes at 1 Hz, and resulting raw data were quality controlled

by using the Hampel filter algorithm for eliminating outliers (Pearson, 2005). The number of particles per unit of volume within each size class ($N(D)$) was computed by dividing the particle volume concentration ($V(D)$) by the diameter ($D$) of a volume-equivalent sphere for the midpoint of each individual class:

$$N(D) = 6\ V(D)\ (\pi\ D^3)^{-1} \tag{1}$$

A total of 25 particle size bins were calculated based on inversions of the scattering pattern and by applying an inversion

kernel matrix derived from scattering patterns of spherical homogenous particles as predicted from Mie theory and a realistic range of index of refraction. The particle size distribution ($N'(D)$) was defined as the average number of particles within a given size class of width $\Delta D$ and per unit of volume (Reynolds et al., 2010):

$$N'(D) = N(D)\ \Delta D^{-1} \tag{2}$$

The parameter $\xi$ was computed as the exponent of the following power-type function:

$$N'(D) = N'(D_o)\ (D/D_o)^{-\xi} \tag{3}$$

where $D_o$ is the reference particle diameter and was set to 35.17 µm. Calculations of $\xi$ were done by least square minimization of log-transformed data (Reynolds et al., 2010). Although particle size distribution in natural waters may not follow a Junge-type slope, its use here was justified since our main interest was to have a first-order assessment of size effects of particulates on optical coefficient's variability. Indeed, the definition of $\xi$ based on LISST measurements applies

for particulates greater than 2 µm. A more realistic representation of PSD is the model proposed by Risovic (1993). This parameterization mainly includes two particle populations ('large' and 'small') having different refractive index and has been recently applied in littoral environments by different studies (Zhang et al., 2013; Zhang et al., 2014; Zhang et al., 2017). Thus, relationships between $\xi$ and optical coefficients in this study are local and should not be generalized to other littoral environments.

**2.5 Optical proxies and particle microphysical characteristics**

The parameter $\gamma$ is positively correlated with the exponent of the particle number size distribution ($\xi = \gamma + 3 - 0.5\ e^{-6\gamma}$, Boss et al., 2001) and negatively related with the mean particle size for particles smaller than 20 µm. The parameter $\gamma$ was derived as the exponent of a power-type regression model of $c_{SPM}$ as a function of wavelength:

$$c_{SPM}(\lambda) = c_{SPM}(488)\ (\lambda/\lambda r)^{-\gamma} \tag{4}$$

where $\lambda r = 488$ nm and it is the reference wavelength (Boss et al., 2013).

The magnitude of *Svis* is positively correlated with extractable iron from crystalline and amorphous iron oxides and organic-iron complexes in measurements corresponding to marine samples (Estapa et al., 2012). Also for the same environments, *Svis* is expected to covary in a direct way with the organic carbon content of particulates (Estapa et al., 2012).

The spectral slope of empirical mass-specific $a_{SPM}$ coefficients (*Svis*) was calculated by nonlinear fitting of a single-exponential decay function over the visible range 400-700 nm:

$$a_{SPM}^{*}(\lambda) = A\ e^{-Svis\,(\lambda\text{-}400)} + B \tag{5}$$

where the term $B$ corresponds to an offset at near-IR wavelengths to account for nonzero absorption by mineral particles (Babin et al. 2003; Röttgers et al., 2014).

### 2.6 Optical cross sections and mass-specific optical coefficients

Spectral values of mass-specific absorption ($\sigma_a^{j}$) and scattering ($\sigma_b^{j}$) cross sections for mineral and organic fractions of SPM were estimated statistically by partitioning each optical coefficient with respect to the concentration of PIM and POM in each sample (see section 2.7). The superscript j indicates PIM or POM chemical fractions. For the case of size fractions of SPM, a mass-specific optical coefficient was calculated for particulate absorption and scattering coefficients:

$$a_i^{*}(\lambda) = a_i(\lambda)\,(m_i)^{-1} \tag{6}$$

$$b_i^{*}(\lambda) = b_i(\lambda)(m_i)^{-1} \tag{7}$$

where m is the mass in g m$^{-3}$ for each size class i.

### 2.7 Statistical analysis

Optical cross sections for chemical fractions of SPM were calculated based on multiple regression model II analysis (i.e., independent and response variables have random errors) (Sokal et al., 1995; Stavn and Richter, 2008):

$$Y = \beta 1\ [CPIM] + \beta 2\ [CPOM] \tag{8}$$

where Y is the response variable representing a specific optical coefficient for unfractionated SPM, $\beta 1$ and $\beta 2$ are partial regression coefficients that correspond with $\sigma^{PIM}$ and $\sigma^{POM}$, respectively. CPIM and CPOM are the concentrations of PIM and POM, respectively, in g m$^{-3}$.

The influence of particle size and chemical composition variations on mass-normalized optical coefficients of particulates ($a_i^{*}$, $b_i^{*}$, $\sigma_a$, $\sigma_b$) and optical proxies ($\gamma$ and *Svis*) was investigated based on correlations with respect to $\xi$ and $F_{SPM}^{PIM}$ variables, respectively. In all cases, the intensity and sign of correlations were quantified based on non-parametric Spearman rank coefficient ($\rho_s$) (Spearman, 1904).

**3 Results**

**3.1 Spatial variability of microphysical properties of SPM**

In terms of particle size distribution, contrasting areas in the SLE-SF were identified. In UE, particulates having a diameter larger than 10 µm had in average a contribution of 11% to the total SPM mass. This proportion was lower in the LE ($F_{SPM}^{>10}$

$^{µm}$ = 0.01-0.11) and SF (0.03-0.15) sub-regions. The largest mass contribution of smallest-sized particulates (i.e., diameter < 0.4 µm) was calculated in the lower estuary ($F_{SPM}^{0.2-0.4\ µm}$ = 0.02-0.27). Lastly, the intermediate size classes 0.4-0.7 µm and 0.7-10 µm were in average the fractions having the largest mass contributions to SPM in SF locations (0.01-0.14 and 0.66-0.87, respectively). In general, the Junge slope calculations suggested the presence of relatively larger particulates in the LE (arithmetic average ± standard deviation = 3.28 ± 0.38, N = 15) with respect to UE (3.46 ± 0.36, N = 3) and SF (3.42 ± 0.39,

N = 5) sub-regions. The uncertainty of ξ calculations, as estimated from 2 standard errors, varied between 1.6 and 10.2% with smaller errors in the LE. Unlike particle size distribution, chemical composition of SPM was less variable throughout the study area ($F_{SPM}^{PIM}$ range = 37 - 87 %). In average, particle composition in UE, SF and LE sub-regions was dominated by minerals ($F_{SPM}^{PIM}$ = 0.65 ± 0.13, 0.67 ± 0.14 and 0.67 ± 0.14 for SF, UE and LE, respectively).

**3.2 Mass-specific optical properties of SPM**

For the spectral interval 400-650 nm, the magnitude of regionally-averaged mass-specific absorption coefficient for unfractioned samples of SPM was higher in SF (e.g., for at λ = 440 nm, arithmetic average ± standard error = 0.523 ± 0.102 $m^2\ g^{-1}$) with respect to UE (0.122 ± 0.068 $m^2\ g^{-1}$) and LE (0.050 ± 0.010 $m^2\ g^{-1}$) locations (Fig. 2a). Conversely, regionally-averaged mass-specific scattering coefficients of unfractionated samples were highly variable within spatial domains even though highest and lowest values tend to be associated with UE (0.499 ± 0.278 $m^2\ g^{-1}$) and LE (0.129 ± 0.046 $m^2\ g^{-1}$)

locations, respectively (Fig. 2b). Size-fractioned mass-specific absorption coefficients tended to be higher in SF (e.g., at λ = 440 nm, up to 2.806 $m^2\ g^{-1}$) with respect to other locations of the SLE (up to 2.111 $m^2\ g^{-1}$) but for the smallest size range 0.2-0.4 µm where some locations belonging to UE (e.g., st 14) showed higher absorption efficiencies per unit of mass (2.187 $m^2$ $g^{-1}$) (Fig. 3a). Spectral curves with the highest $a_i$* values (e.g., up to 4 $m^{-1}$ at λ = 400 nm) corresponded with the smallest-sized and largest-sized fractions of SPM (Fig. 3a,d). These values were up to 8 and 5 times higher than those characteristic of

size fractions 0.4-0.7 µm and 0.7-10 µm, respectively (Fig. 3b-c). Similar to $a_i$*, highest $b_i$* values (up to 5.7 $m^2\ g^{-1}$ for λ = 400 nm) corresponded with size fractions having particles with the smallest and the largest diameter (Fig. 4). In general, the spectral slope of $b_i$* was very variable in all size fractions (-6 $10^{-5}$ to 6.28 $10^{-3}$ $nm^{-1}$) with the greatest spectral changes associated to particulates greater than 10 µm. Highest scattering efficiencies in terms of $b_i$* were not always measured in the same region. Indeed, maximum $b_i$* values for size fraction 0.7-10 µm (up to 1.246 $m^2\ g^{-1}$ at λ = 556 nm) and >10 µm (up to

4.579 $m^2\ g^{-1}$) were obtained in UE and LE domains, respectively. A common finding was the larger magnitude of size-fractionated mass-specific particulate absorption and scattering coefficients with respect to true optical cross sections of

chemical fractions (up to 2 and 3 orders of magnitude for total absorption and scattering, respectively) (Fig. 5). To exemplify these differences, the range of $a_{0.2\text{-}0.4\,\mu m}^*$, $a_{>10\,\mu m}^*$, $\sigma_a^{PIM}$ and $\sigma_a^{POM}$ values measured at a wavelength of 440 nm and over the whole study area was 0.05-2.14, 0.18-1.20, 0.01-1.06 and 0.01-1.03 m$^2$ g$^{-1}$, respectively (Fig. 5a). Likewise, for a wavelength of 556 nm, the range of $b_{0.2\text{-}0.4\,\mu m}^*$, $b_{>10\,\mu m}^*$, $\sigma_b^{PIM}$ and $\sigma_b^{POM}$ values was 1.82-2.39, 1.05-1.49, 0.08-0.36 and 0.07-0.38 m$^2$ g$^{-1}$,

respectively (Fig. 5b). In general for the spectral range of 440-556 nm, empirical mass-specific absorption coefficients tended to be higher for particulates within the lower size range (i.e., 0.2-0.4 µm) (Fig. 5a, left-axis). Also, this trend appeared to be reversed at longer wavelengths. Unlike mass-specific absorption coefficients of size fractions, true optical cross sections of chemical fractions showed only differences within the red and near-IR wavelengths (Fig. 5a, right-axis). For the whole study area, the arithmetic average of mass-specific scattering coefficients for the size fraction 0.2-0.4 µm were larger

with respect to that associated to the size fraction >10 µm (Fig. 5b, left-axis). At a wavelength of 440 nm, the true scattering cross sections for PIM were substantially higher (1.060 ± 0.206 m$^2$ g$^{-1}$) than those corresponding to POM (0.359 ± 0.123 m$^2$ g$^{-1}$) (Fig. 5b, right-axis). The spatial variation of mass-specific coefficients and true optical cross sections of different fractions of SPM are depicted in Fig. 6. Notice that true absorption or scattering cross sections for chemical fractions of SPM are not shown in UE locations given the insufficient number of samples to perform a multiple regression analysis. In

Saguenay Fjord waters, the maximum $a_{SPM}^*(440)$ values (up to 4.6 m$^2$ g$^{-1}$) were associated with the size fraction of SPM having particulates greater than 10 µm (Fig. 6a, left-axis). Unlike mass-specific absorption coefficients of SPM size fractions, no substantial sub-regional differences were detected for $\sigma_a^{PIM}(440)$ and $\sigma_a^{POM}(440)$ values ($P > 0.05$, $t$ up to 11.5, Student-t test) (Fig. 6a, right-axis). In general, $\xi$ and $F_{SPM}^{PIM}$ correlations with size-fractionated mass-specific optical coefficients suggest that particle chemical composition has a larger influence on $a_i^*(440)$ ($\rho_s$ up to 0.50, $P = 0.0009$) with

respect to particle size ($\rho_s$ up to 0.32, $P = 0.0033$) (Table 2). The regional average of $b_i^*(550)$ in UE-SF (0.432-0.501 m$^2$ g$^{-1}$) was larger with respect to that computed in LE waters (0.136 ± 0.027 m$^2$ g$^{-1}$) only for particulates within the size range 0.7-10 µm (Fig. 6b, left-axis). Also for SPM fraction having the largest particulates (i.e., > 10 µm), UE locations had typically larger $b_i^*(550)$ values with respect to SF-LE regions. In general and unlike $b_i^*$, no clear sub-regional differences were observed between $\sigma_b^{PIM}(440)$ and $\sigma_b^{POM}(440)$ values ($P > 0.05$, $t$ up to 13.2, Student-t test) (Fig. 6b, right-axis). Unlike

$a_i^*(440)$, $b_i^*(550)$ variability was less influenced by changes on particle composition ($\rho_s$ up to 0.42, $P = 0.0015$) (Table 2). Conversely, the impact of changing particle dimensions, as inferred from $\rho_s$ correlations, was greater for $b_i^*(550)$ ($\rho_s$ up to 0.37, $P = 0.006$) with respect to $a_i^*(440)$ ($\rho_s$ up to 0.32, $P = 0.009$) values.

### 3.4 Optical proxies

Correlations between mass contributions to different size and chemical fractions of SPM and optical proxies are presented in

Table 3. Over the whole study area, there was not a clear relationship between $\gamma$ and chemical fractions of SPM fractions ($\rho_s = -0.34$, $P = 0.11$). However, $\gamma$ responded to variations on size fractions for the range 0.2-10 µm ($\rho_s$ up to 0.53, $P = 0.01$). The sign of the relationship changed depending on the size class under investigation (positive for small-sized, negative for

intermediate-sized particulates). Although positively correlated, there was not a clear relationship between $\gamma$ and $\xi$ determinations ($\rho_s = 0.15$, $P = 0.49$, N = 23). The range of $\gamma$ values was 0.759-3.282, 1.389-1.534, 2.873-3.282 and 0.759-1.802 nm$^{-1}$ for the SLE, UE, SF and UE domains. The uncertainty of $\gamma$ determinations varied between 2.2% and 6.4% with largest errors for samples obtained in LE waters. The spectra slope of $a_{SPM}^{*}$ was not substantially affected by $F_{SPM}^{PIM}$

changes ($\rho_s = -0.15$, $P = 0.49$, N = 23), however *Svis* variability was strongly influenced by particle size changes within the interval 0.2-0.7 $\mu$m ($\rho_s = -0.49$, $P = 0.008$). Range of *Svis* values of unfractionated samples was 0.005-0.051, 0.009-0.017, 0.014-0.051 and 0.005-0.016 nm$^{-1}$ for the SLE, UE, SF and UE domains, respectively. The uncertainty of *Svis* estimates varied between 0.5 and 21.5% with largest errors corresponding with samples obtained in LE locations. Over the whole study area, the range of *Svis* values was 0.004-0.026, 0.007-0.052, 0.004-0.109 and 0.001-0.028 nm$^{-1}$ for size fractions 0.2-

0.4 $\mu$m, 0.4-0.7 $\mu$m, 0.7-10 $\mu$m and > 10 $\mu$m, respectively. In general, *Svis* slopes were not correlated between size fractions even though the magnitude of *Svis* for total unfractioned samples was strongly influenced by *Svis* calculated for the 0.7-10 $\mu$ fraction ($\rho_s = 0.66$, $P = 0.004$).

## 4 Discussion

### 4.1 Uncertainty of optical measurements

Inherent optical properties in this study were derived from an ac-s instrument. Thus, large errors on absorption coefficients may be anticipated in relatively turbid waters if original measurements are not corrected by scattering effects (Boss et al., 2009; McKee et al., 2013). These effects are mainly attributed the acceptance angle of the transmissometer and the multiple scattering of photons. The acceptance angle of the ac-s instrument is ~0.9° and much larger than that corresponding to the LISST-100X diffractometer (~0.027°). Thus, a larger underestimation on $c$ magnitude is expected in ac-s with respect to

LISST-100X measurements due to a larger contribution of forward-scattered photons arriving to the detector of the former optical instrument. Further comparisons of $c$(532) measurements derived here by ac-s and LISST-100X showed that $c$ values as derived from ac-s were 23-84% lower with respect to those determinations based on LISST-100X. This is consistent with Boss et al. (2009) who reported that uncorrected Wet Labs ac-9 attenuation values are approximately 50%-80% of equivalent LISST attenuation data. Unfortunately, $c$ deviations due to acceptance angle variations were not corrected in this study due

to the lack of true $c$ values as obtained by using an integrating cavity absorption meter (e.g., PSICAM) (Röttgers et al., 2005). Notice that these errors are much greater with respect to the optical variability associated to each sample determination in SLE-SF waters and computed based on ac-s measurements (e.g., < 1% at $\lambda = 532$ nm).

In this investigation, the 'flat' baseline correction was selected for correcting residual scattering in absorption coefficient estimates as derived from ac-s measurements. This technique was chosen due to the lack of PSICAM measurements or

critical ancillary optical information (e.g., particle backscattering efficiency) to tune up a Monte Carlo scattering correction approach (McKee et al., 2008). The 'flat' scattering correction approach is expected to provide a fair correction of $a$ values

in oceanic waters (up to 15% underestimation at wavelengths shorter than 600 nm, see Fig. 8b, McKnee et al., 2013) but may result in large deviations (up to 100% decrease in the NIR) of $a$ values in relatively turbid waters (e.g., $a > 0.2$ m$^{-1}$) such as the Baltic/North Sea. Also, this issue is present when the proportional correction method of Zaneveld et al. (1994) is applied. Unlike the 'flat' baseline, the scattering residual of the proportional method is spectrally dependent but still relying in one

reference wavelength in the NIR spectral range. Approximations justifying the use of the 'flat' (i.e., zero absorption signal in the NIR) and 'proportional' (i.e., wavelength-dependent scattering phase function) method are still in debate (McKnee et al., 2013). Lastly, the Monte Carlo correction method (McKee et al., 2008) has in general better agreement (error <10%) with true $a$ values as derived from an integrating cavity absorption meter. However, this approach may also have major uncertainties due to assumptions regarding optical coefficients (e.g., particulate backscattering ratio and volume scattering

function) and changes on scattering efficiency by the inner wall of the reflective tube due to aging (McKnee et al., 2013). Thus in conclusion, the resulting optical coefficients and mass-specific optical coefficients of particulates measured in SLE-SF waters may present large errors (i.e., > 50%) with respect to true values and at wavelengths longer than 550 nm. This bias is anticipated to be maximum (minimum) in UE (LE) locations.

### 4.2 Variability of microphysical properties of SPM

A striking finding in this study was the important weight contribution of relatively large particulates (i.e., >10 µm) in UE waters. This phenomenon was likely attributed to the active resuspension of sediments associated with vertical mixing produced by tidal currents and winds (Yeats, 1988). Conversely, this effect was secondary in relatively deep waters of SF and LE where large and heavy particulates are rapidly removed from the water column and deposited along submarine canyons (Gagné et al., 2009). Although chemical composition of size-fractioned SPM was not analyzed in this study,

additional correlations with $F_{SPM}^{PIM}$ suggest that particulates smaller than 10 µm were richer in inorganic matter ($\rho_s = 0.62$, $P < 0.001$, N = 23) with respect to particulates with a diameter greater than 10 µm. This finding confirms previous studies showing that relatively small (~2 µm) particulates in the SLE are mainly composed by minerals (Yeats, 1988; Gagné et al., 2009). In this contribution, a large proportion of particulates with a diameter above 50 µm and lower $\xi$ values were typically found in LE locations. These results also support historical observations made during July and August and showing a greater

proportion of relatively large particulates (i.e., > 5 and < 50 µm) over the LE locations (Chanut and Poulet, 1979).

### 4.3 Spatial variability of mass-specific optical coefficients

In this study, $a_{SPM}^*$ measurements in the visible and near-IR range had a large variability that was comparable to the range of values reported in the literature for temperate coastal waters (e.g., Mobile Bay, River of La Plata, Elbe Estuary, Gironde Estuary) (Stavn and Richter, 2008; Doxaran et al., 2009; Dogliotti et al., 2015) (Table 4). In general, lowest $a_{SPM}^*$ values

(i.e., 0.01-0.02 m$^2$ g$^{-1}$ at $\lambda = 440$ nm) commonly corresponded with samples obtained in very turbid environments (i.e., > 100 g m$^{-3}$, Mississippi River and Delta, Gironde River, La Plata River) (Bowers and Binding, 2006; D'Sa et al. 2006; Dogliotti et

al., 2015; Doxaran et al., 2009). Notice that part of this decrease can be attributed to an incomplete removal of multiple scattering effects. Relative low $a_{SPM}^*$ values have been linked to high POC/SPM (Wozniak et al., 2010) and chl/SPM concentration ratios, where chl means chlorophyll a (Estapa et al., 2012). In this study, chl/SPM presented values as high as $2 \times 10^{-3}$ that are comparable to relatively high ratios reported by D'Sa et al. (2006). Thus, it is suggested that some locations in our study area are characterized by relatively high POC/SPM as other turbid coastal environments such as adjacent waters to the Mississippi Delta (D'Sa et al. 2006).

A well-known mechanism explaining the general decrease of $a_{SPM}^*$ in very turbid waters is related to packaging effects (Morel, 1974; Zhang et al., 2014). At higher turbidities, larger particulates contribute to PSD variations, thus as mean diameter of particles increases, the light absorption efficiency per averaged particle decreases (i.e., the interior of larger particles has a greater 'shading'). This could also explain the spatial differences of $a_{SPM}^*(440)$ in our study area where larger values corresponded with surface waters dominated by particles assemblages having a smaller mean diameter (i.e., UE and SF). In nearshore waters of California, Wozniak et al. (2010) demonstrated inverse relationships between $a_{SPM}^*(440)$ and the median particle diameter for inorganic- and organic-dominated assemblages. Also and consistent with our previous discussion regarding particle composition, Wozniak et al. (2010) observed that POC/SPM was positively correlated with the median particle diameter. Indirect size effects on $a_{SPM}^*(440)$ due to changes on iron content per particle have been discussed by Estapa et al. (2012). In general, smaller particulates have a greater surface for adsorbing organic compounds where iron can accumulate (Mayer, 1994; Poulton and Raiswell, 2005). Thus, SPM fractions with smaller particulates are expected to have an enhancement of $a_{SPM}^*(440)$ due to high iron concentrations. This phenomenon likely explained our higher $a_{SPM}^*(440)$ in SF regions with respect to LE waters where the water salinity range is 0-29 and 29-33.5, respectively (El Sabh, 1988). Indeed, relatively high concentrations of particulate iron have been measured in surface waters of the Saguenay Fjord (Yeats and Bewers, 1976; Tremblay and Gagné, 2009). In coastal Louisiana and the lower Mississippi and Atchafalaya rivers, Estapa et al. (2012) found that magnitude of $a_{SPM}^*$ within the UV ($\lambda \sim 360\text{-}390$ nm) and blue ($\lambda \sim 400\text{-}450$ nm) spectral range is commonly higher in freshwater with respect to marine samples. This is related to the greater concentration of particulate iron oxides and hydroxides derived from terrestrial sources in freshwater samples and later transport and reduction in marine environments. Iron oxide and hydroxide minerals have a major light absorption within the spectral range of 400-450 nm due to the absorption bands of iron (Estapa et al., 2012). Pigmentation of mineral particulates due to iron hydroxides has been suggested to be a major factor increasing $a_{SPM}^*$ (Babin and Stramski, 2004; Estapa et al., 2012). Unfortunately and unlike optical measurements made by Estapa et al. (2012), the resolution of our ac-s measurements (~4 nm) did not allow a deeper analysis of iron absorption peaks by performing a second-derivative calculation. In general, $\sigma_a^{POM}$ and $\sigma_a^{PIM}$ values were within the range of values reported in the literature with the exception of SF locations where mass-specific absorption cross sections were substantially higher (up to 1.71 and 0.86 $m^2\,g^{-1}$, respectively, $\lambda = 440$ nm). This difference was likely attributed to the aforementioned enhancement of light absorption due to particulate iron-enrichment in SF waters.

Similar to $a_{SPM}^*$, $b_{SPM}^*$ values were highly variable between locations and within the range of measurements obtained in other environments (Table 4). In this study, the spectral variation $b_{SPM}^*$ between regions showed a spectral flattening as particle assemblages become dominated by organic matter (i.e., LE). This finding is consistent with Wozniak et al. (2010) measurements made in Imperial Beach, California. Our measurements of scattering cross sections of PIM in the SLE were

higher with respect to other littoral regions of the world. For instances, $\sigma_b^{PIM}(440)$ in the SLE was up to 2-fold the magnitude of maximum $\sigma_b^{PIM}(440)$ values measured in off New Jersey coast by Snyder et al. (2008). The origin of these differences is unknown and could be mainly related to mineral composition variations and associated iron as particle size distribution measurements during our surveys were comparable to those published by other studies. Unlike $\sigma_b^{PIM}$, our $\sigma_b^{POM}$ estimates were within the range of values obtained in the Gulf of Mexico and along the east coast of US.

**4.4 Particle size and chemical composition effects on mass-specific optical coefficients**

Correlations of ξ and $F_{SPM}^{PIM}$ with mass-specific optical coefficients for different SPM size fractions were shown in Table 2. For all size fractions, ξ was positively correlated with $a_i^*(440)$ ($\rho_s$ up to 0.32, $P = 0.006$). This pattern is consistent with the higher absorption efficiency of relatively small-sized particulates. As previously discussed, these particulates have a greater light absorption per unit of particle mass due to iron-enrichment and a lesser role of shading effects. Since particle

aggregates were altered during our experiments, the influence of particle density on mass-specific optical coefficients cannot be quantified as this effect is mainly observed in undisrupted marine aggregates (Slade et al. 2011; Neukermans et al., 2012, Neukermans et al 2016). However and based on Estapa et al. (2012) simulations, the impact of aggregation on $a_{SPM}^*$ is anticipated to be small (i.e., ~10%) with respect to the spatial variability of $a_{SPM}^*$ in SLE-SF waters.

In general, ξ was positively correlated with $b_i^*(550)$ ($\rho_s$ up to 0.37, $P = 0.008$) and pointed out as expected the higher

scattering efficiency of small-sized particulates due to the smaller influence of packaging effects. Notice that ξ correlations with $b_i^*(550)$ were greater with respect to $a_i^*(440)$ and more remarkable for relatively large-sized particulates. In Arctic waters, Reynolds et al. (2016) observed an increase on mass-specific particulate backscattering for mineral-rich particle assemblages that tend to exhibit steeper size distributions. Although no particulate backscattering measurements were available in this study, Reynolds et al. (2016) highlight the importance of relatively small-sized particulates for driving

variations on mass-specific optical coefficients linked to scattering processes.

In all cases, $F_{SPM}^{PIM}$ had a stronger correlation with $a_i^*(440)$ compared with $b_i^*(550)$ values, and these relationships were stronger when SPM was dominated by particulates with an intermediate size (i.e., 0.4-10 μm). The enrichment of suspended particulates on inorganic matter and concomitant variations $a_i^*(440)$ may be explained by a greater contribution of mineral-associated iron to light absorption. Also, the combustion method used to measure PIM in our study could be another factor

explaining the increased particle absorption in the blue range (Babin et al. 2003). Iron can take many forms in mineral particulates (oxides, hydroxides, monosulfides) and can be deposited over the particle surface or be part of its internal structure (e.g., clays). Since the mean diameter of clay particles is less than 2 μm, the aforementioned $F_{SPM}^{PIM}$ -$a_i^*(440)$

correlations were also likely affected by iron associated (adsorbed or structural) to other types of inorganic particulates that are characterized by larger dimensions. In SF locations, reduced iron is mainly associated to dissolved organic compounds that can be strongly adsorbed to hydrous metal oxides (Deflandre et al., 2002). Babin and Stramski (2004) obtained positive correlations between $a_{SPM}^*$ and iron content of dust and soil particles suspended in seawater. Estapa et al. (2012) found a

strong covariation between $a_{SPM}^*$ values and dithionite-extractable iron content of oxides and hydroxides.

An important objection to correlations of $\xi$ and $F_{SPM}^{PIM}$ with mass-specific optical coefficients of SPM size fractions was related to differences in terms of particle size range used to compute $\xi$ and $F_{SPM}^{PIM}$ and particle size classes derived by sequential filtration of water samples. More specifically, $\xi$ is not representative of submicron particles less than 2 μm. Also, $F_{SPM}^{PIM}$ is only a valid particle composition parameter for particles mostly larger than 0.7 μm. Thus, correlations $\xi$ and

$F_{SPM}^{PIM}$ with mass-specific optical coefficients of 0.2-0.4 μm and 0.4-0.7 μm may only reflect indirect dependencies between mass-normalized optical coefficients of different size classes. This possibility (i.e., correlations between $a_i^*$ or $b_i^*$ of different size classes) was confirmed based on samples obtained in UE, LE and SF waters. Lastly, it is important to discuss the potential bias on $a_i^*$ and $b_i^*$ determinations due to size fractionation and *a posteriori* impact on correlations with respect to $F_{SPM}^{PIM}$ and $\xi$. No measurements of $F_{SPM}^{PIM}$ and $\xi$ were done in size fractions of SPM, thus it is difficult to compare

particulate size distribution and chemical composition changes before and after the size fractionation of the samples. Size fractionation is anticipated to cause retention of smaller particulates in membranes having a larger pore size. These primary particles will overestimate the weight of the filtered sample and underestimate the weight of the next filtration step consisting in a membrane having a smaller pore size. Since particle sieving begins with large-sized particles and finishes with small-sized particles, the magnitude of $a_i^*$ and $b_i^*$ for relatively large (small) particulates is likely to be under-(over-)

estimated. Bias on mass of size fractions was verified by comparing the sum of masses for 0.7-10 μm and >10 μm with the total sample filtered trough a GF/F filter (i.e., 0.7 μm nominal pore size). The arithmetic average of relative bias for the whole study area was 31.4% or a 31.4% overestimation of mass for particulates > 0.7 μm when total weight is computed based on sum of partial weights corresponding to different size fractions. An optimization scheme to adjust the mass for each size fractions (i.e. adjusting the various masses to sum up to the total mass filtered) was not attempted since we didn't filter

total samples through 0.2 or 0.4 μm membranes due to the sequential mode of our filtration. Thus, 'filtration weighting factors' for size fractions > 0.2 μm or > 0.4 μm could not be calculated.

### 4.5 Optical proxies of particle characteristics

In terms of fractioned mass, the size of particulates was the dominant variable driving changes on $\gamma$ ($\rho_s$ up to 0.53, $P$ = 0.004). Conversely, the mineral content of SPM did not have a statistically detectable impact at 95% confidence interval. In

particular, the strongest response of $\gamma$ to size effects was manifested for the mass fraction having the smallest particulates (i.e., 0.2-0.4 μm). Despite the major effects of particle size classes on $\gamma$, values of $\gamma$ were not clearly correlated with $\xi$ slopes. In oceanic waters, $\xi$ and $\gamma$ values are expected to covary in a linear way for a specific range of refractive index and $\xi$ (Boss et

al., 2001; Twardowski et al., 2001). Our range of $\xi$ values was within the natural variability reported in coastal and oceanic environments ($\xi$ = 2-4.5) (Reynolds et al., 2010; Neukermans et al., 2012; Xi et al., 2014). Also, the magnitude of $\gamma$ in our samples (0.29-2.22 $nm^{-1}$) was within the range of values that characterize oceanic environments (0.2-2) (Twardowski et al., 2001, Boss et al., 2013). Unlike oceanic waters, the poor correspondence between $\xi$ and $\gamma$ values in this study was linked to

different responses of spectral $c_{SPM}$ and particle size distribution slopes to changes of two non-covarying optical contributions: minerals and phytoplankton. Also, the reduced number of sampling locations and the geographic variability of $\xi$-$\gamma$ relationships were additional factors likely explaining the lack of a general functionality for the study area. Lastly, $\xi$ and $\gamma$ were not substantially correlated in our samples due to deviations on Mie-based models (e.g, absorbing spheres) of $\gamma$ as a function of $\xi$ (Twardowski et al., 2001). Indeed during our surveys, high absorbing particulates were present in SLE-SF

waters.

The variability of *Svis* values in this study was relatively high (~10-fold) with respect to other littoral environments (1.3-fold, *Svis* = 0.009-0.0113 $nm^{-1}$) (Estapa et al., 2012). Since *Svis* was preferentially influenced in a direct way by the contribution of small-sized particulates within the range 0.2-0.4 $\mu$m, it is feasible a potential link between *Svis* and particulate iron of small-sized mineral particulates (Estapa et al., 2012). No statistically significant correlations at 95% confidence level were

computed between $F_{SPM}^{PIM}$ and *Svis*. This is counterintuitive as $F_{SPM}^{PIM}$ is positively related to $a_i^*$ and presumably iron content of particulates. This discrepancy might be related to the inclusion of freshwater or brackish samples into the correlation analysis as *Svis* is only expected to change with extractable-iron of marine measurements (Estapa et al., 2012). More specific correlations by only using LE measurements supported this hypothesis ($\rho_s$ = 0.58, *P* = 0.023). Thus, our results suggest that *Svis* is likely an indicator of iron associated to mineral-enriched particulates in LE waters.

**5 Conclusions**

The measure of optical cross sections of SPM is essential for developing optical inversions and improves our understanding regarding the origin of optical signatures in remote sensing studies and map biogeo-chemical components in surface waters. In this contribution, we presented for the first time, mass-specific scattering and absorption coefficients of size fractioned SPM in estuarine waters of the Saint Lawrence River and a major SLE tributary, the Saguenay Fjord.

Despite the intrinsic variability of weight-normalized optical coefficients due to variations of particle micro-physical attributes, the following patterns were identified: 1. the mass-specific absorption coefficient of SPM was preferentially influenced by changes in particle chemical composition as inferred from changes on $F_{SPM}^{PIM}$, 2. regional variations on *Svis* suggest a substantial iron-enrichment of suspended particulates in LE waters, 3. $a_{SPM}^*$(440) values were usually higher in SF-UE with respect to LE locations for all size fractions and indicate that iron is not selectively bounded to specific size class of

particulates, 4. *Svis*- $F_{SPM}^{PIM}$ correlations in LE locations suggest a potential iron-enrichment of particulates having a larger mineral content, 5. salinity was an important variable correlated with changes on $a_{SPM}^*$ at the regional scale, 6. size spectra of particulates had a larger impact on $b_{SPM}^*$ than $a_{SPM}^*$, and 7. no clear regional differences were established in terms of $b_{SPM}^*$

magnitude or spectral variation. In summary, the aforementioned relationships will be useful in investigating local and regionally-limited relationships and properties of SPM.

**6 Funding**

This investigation was supported by the Natural Sciences and Engineering Research Council of Canada, Individual Discovery grant, project title: "Optical remote Sensing models of suspended Particulate matter in the St. Lawrence Estuary "(OSPLE), awarded to Dr. Martin Montes Hugo.

**7 Acknowledgements**

We thank to the crew of the Creed and Mr. Alexandre Palardy for their assistance during the field work. Also, we appreciate the support of ISMER technicians Mr. Pascal Rioux and Ms. Dominique Lavallée during the field surveys and the processing of lab measurements.

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

**Table 1. Summary of acronyms**

| Abbreviation | Definition | Unit |
|---|---|---|
| SLE | St. Lawrence Estuary | |
| UE | Upper Estuary | |
| SF | Saguenay Fjord | |
| LE | Lower Estuary | |
| CSPM | Concentration of suspended particulate matter | $g\,m^{-3}$ |
| $F_{SPM}^{i}$ | Contribution of size fraction i to total mass of SPM | dimensionless |
| $F_{SPM}^{j}$ | Contribution of chemical fraction j to total mass of SPM | dimensionless |
| NAP | Non-algal particulates | |
| CDOM | Chromophoric dissolved organic matter | |
| PIM | Particulate inorganic matter | $g\,m^{-3}$ |
| POM | Particulate organic matter | $g\,m^{-3}$ |
| $\lambda$ | Light wavelength | nm |
| $a_{SPM}$ | Absorption coefficient of SPM | $m^{-1}$ |
| $b_{SPM}$ | Scattering coefficient of SPM | $m^{-1}$ |
| $a_{SPM}^{*}$ | Mass-specific absorption coefficient of SPM | $m^2\,g^{-1}$ |
| $b_{SPM}^{*}$ | Mass-specific scattering coefficient of SPM | $m^2\,g^{-1}$ |
| $\sigma_{a}^{j}$ | Absorption cross section of SPM chemical fraction j | $m^2\,g^{-1}$ |
| $\sigma_{b}^{j}$ | Scattering cross section of SPM chemical fraction j | $m^2\,g^{-1}$ |

| | | |
|---|---|---|
| ξ | Differential Junge slope of particle size distribution | Number of particulates per µm |
| D | Diameter of a volume-equivalent sphere at mid point of size class | µm |
| V(D) | Volume concentration at size class D | µL L$^{-1}$ |
| N(D) | Particle number concentration at size class D | m$^{-3}$ |
| N'(D) | Particle number density at size class D | m$^{-3}$ µm$^{-1}$ |
| γ | Spectral slope of particulate beam attenuation coefficient | nm$^{-1}$ |
| *Svis* | Spectral slope of mass-specific particulate absorption coefficient within the visible spectral range | nm$^{-1}$ |

| | | |
|---|---|---|
| ξ | Differential Junge slope of particle size distribution | Number of particulates per µm |
| D | Diameter of a volume-equivalent sphere at mid point of size class | µm |
| V(D) | Volume concentration at size class D | µL L$^{-1}$ |
| N(D) | Particle number concentration at size class D | m$^{-3}$ |
| N'(D) | Particle number density at size class D | m$^{-3}$ µm$^{-1}$ |
| γ | Spectral slope of particulate beam attenuation coefficient | nm$^{-1}$ |
| *Svis* | Spectral slope of mass-specific particulate absorption coefficient | nm$^{-1}$ |

**Table 2. Particle size and chemical composition effects on mass-specific optical coefficients. Spearman rank correlations for $a_i^*$ and $b_i^*$ are computed at a wavelength of 440 and 550 nm, respectively.**

| Mass-specific Optical fraction | $\xi$ | $F_{SPM}^{PIM}$ |
|---|---|---|
| $a_{0.2-0.4\,\mu m}^*$ | 0.32 * | 0.31 * |
| $a_{0.4-0.7\,\mu m}^*$ | 0.28 * | 0.50 ** |
| $a_{0.7-10\,\mu m}^*$ | 0.26 * | 0.49 * |
| $a_{>10\,\mu m}^*$ | 0.31 * | 0.44 * |
| $b_{0.2-0.4\,\mu m}^*$ | 0.15 | -0.17 * |
| $b_{0.4-0.7\,\mu m}^*$ | 0.05 | -0.06 |
| $b_{0.7-10\,\mu m}^*$ | 0.23 * | 0.42 * |
| $b_{>10\,\mu m}^*$ | 0.37 * | 0.26 * |

**Table 3. Correlation of optical proxies with particle size and composition. Spearman rank correlations based on 23 samples.**

| Mass fraction of particulates | $\gamma$ | $Svis$ |
|---|---|---|
| $F_{SPM}^{PIM}$ | -0.34 | -0.15 |
| $F_{SPM}^{0.2\text{-}0.4\ \mu m}$ | 0.53* | 0.49** |
| $F_{SPM}^{0.4\text{-}0.7\ \mu m}$ | -0.43* | -0.49** |
| $F_{SPM}^{0.7\text{-}10\ \mu m}$ | -0.38* | -0.30* |
| $F_{SPM}^{>10\ \mu m}$ | 0.13 | 0.19 |

**Table 4. Mass-specific optical coefficients of suspended particulates for different littoral environments. Acronyms are defined in Table 1.**

| Location | $\lambda$ | $a_{\text{SPM}}^{*}$ | $b_{\text{SPM}}^{*}$ | $\sigma_a^{\text{POM}}$ | $\sigma_a^{\text{PIM}}$ | $\sigma_b^{\text{POM}}$ | $\sigma_b^{\text{PIM}}$ | CSPM | References |
|---|---|---|---|---|---|---|---|---|---|
| UE | 440 | 0.01 – 0.25 | 0.01 – 1.06 | 0.15 | 0.11 | 0.84 | 2.27 | 7.38 – 30.6 | This study |
| | 488 | 0.01 – 0.14 | 0.01 – 0.97 | 0.06 | 0.05 | 0.76 | 2.04 | | |
| | 556 | 0.01 – 0.06 | 0.01 – 0.86 | 0.01 | 0.01 | 0.71 | 1.82 | | |
| | 665 | 0.01 – 0.02 | 0.01 – 0.73 | 0.01 | 0.05 | 0.45 | 1.67 | | |
| | 708 | 0.01 – 0.012 | 0.01 – 0.68 | 0.01 | 0.02 | 0.11 | 1.31 | | |
| SF | 440 | 0.32 - 0.73 | 0.20-0.56 | 1.71 | 0.86 | 1.78 | 0.94 | 2.28 – 3.68 | |
| | 488 | 0.17 - 0.39 | 0.18-0.49 | 1.84 | 0.43 | 1.14 | 0.88 | | |
| | 556 | 0.08 – 0.17 | 0.15-0.42 | 0.85 | 0.17 | 0.45 | 0.56 | | |
| | 665 | 0.02 – 0.04 | 0.13 – 0.34 | 0.12 | 0.11 | 0.23 | 0.12 | | |
| | 708 | 0.01 – 0.02 | 0.12 – 0.31 | 0.01 | 0.01 | 0.12 | 0.04 | | |
| LE | 440 | 0.03 – 0.07 | 0.04 – 0.22 | 0.07 | 0.02 | 2.64 | 2.04 | 2.72 – 25.7 | |
| | 488 | 0.02 – 0.04 | 0.04 – 0.21 | 0.03 | 0.01 | 2.13 | 1.88 | | |
| | 556 | 0.01 – 0.02 | 0.04 – 0.19 | 0.01 | 0.01 | 1.88 | 1.36 | | |
| | 665 | 0.003 – 0.006 | 0.04 – 0.18 | 0.02 | 0.01 | 1.42 | 0.89 | | |
| | 708 | 0.015 – 0.002 | 0.04 – 0.17 | 0.02 | 0.01 | 0.98 | 0.67 | | |

| Location | $\lambda$ | $a_{\text{SPM}}^{*}$ | $b_{\text{SPM}}^{*}$ | $\sigma_a^{\text{POM}}$ | $\sigma_a^{\text{PIM}}$ | $\sigma_b^{\text{POM}}$ | $\sigma_b^{\text{PIM}}$ | CSPM | References |
|---|---|---|---|---|---|---|---|---|---|

| Location | λ | | | | | | Reference |
|---|---|---|---|---|---|---|---|
| Elber River, | 650 | 0.001 – 0.020 | – | | | 0.5-10 | Röttgers et al. (2014) |
| German Bight, | 750 | 0.001 – 0.019 | – | | | | |
| Baltic Sea, New Caledonia lagoon | 850 | 0.001 – 0.014 | – | | | | |
| Monterey Bay, US | 532 | | 0.46 – 2.54 | 1.23–3.39 | 0.08 – 0.77 | 0.11 – 2.37 | Zhang et al. (2014) |
| Mobile Bay, US | 532 | | 0.40 – 1.78 | 0.35–3.85 | 0.27 – 0.79 | 0.26 – 7.36 | |
| Hudson Bay, Canada | 675 | 0.001 – 0.12 | | | | 0.2 – 2.5 | Xi et al. (2013) |
| Mississippi River, US | 450 | 0.02 – 0.11 | | | | 7-25 | Bowers and Binding (2006) |
| | 550 | 0.017 – 0.06 | | | | | |
| | 650 | 0.012–0.035 | | | | | |
| | 700 | 0.01 – 0.025 | | | | | |

| Location | λ | | | | | | | Reference |
|---|---|---|---|---|---|---|---|---|
| Mobile Bay, | 440 | 0.44 – 1.95 | | | 0.01-1.91 | 0.36 – 0.80 | 0.23-25.32 | Stavn and Richter (2008) |
| Southwest Pass, US | 488 | 0.41 – 1.89 | | | 0.01-1.82 | 0.36-0.73 | | |
| | 550 | 0.40 – 1.80 | | | 0.01-1.65 | 0.33-0.70 | | |
| | 676 | 0.36 – 1.63 | | | 0.04-1.48 | 0.34-0.63 | | |
| | 715 | 0.34 – 1.61 | | | 0.02-1.39 | 0.33-0.58 | | |
| Coast of New Jersey, | 440 | | 0.23 – 0.59 | 0.08– 0.17 | 0.7 – 5.1 | 0.3 – 1.3 | 0.44 – 6.6 | Snyder et al. (2008) |
| Monterey Bay, | 488 | | 0.18 – 0.39 | 0.07– 0.13 | 0.65 – 4.8 | 0.4 – 1.6 | | |
| Great Bay | 556 | | 0.13 – 0.21 | 0.05– 0.08 | 0.4 – 4.3 | 0.5 – 1.8 | | |
| Mobile Bay | 665 | 0.05 ± 0.01 (arithmetic mean ± standard deviation) | 0.09 – 0.11 | 0.05– 0.06 | 0.35 – 3.8 | 0.4 – 1.7 | | |
| | 708 | | 0.02 – 0.03 | 0.01– 0.02 | 0.4-3.9 | 0.3-1.7 | | |
| Irish sea, UK | 665 | | 0.08 – 0.45 | 0.01 – 0.02 | | 0.47 – 0.49 | 1.9 – 26.5 | Binding et al. (2005) |

| Location | | | | | | | |
|---|---|---|---|---|---|---|---|
| Irish sea, UK | 443 | 0.062 ± 0.013 | 0.17 – 0.19 | 0.05 – 0.06 | 0.25 – 0.27 | 1.6 – 50 | Bowers and Binding (2006) |
| | 490 | | 0.20 – 0.22 | 0.03 – 0.04 | 0.33 – 0.37 | | |
| | 555 | | 0.20 – 0.24 | 0.03 – 0.03 | 0.37 – 0.39 | | |
| | 665 | | 0.14 – 0.15 | 0.02 – 0.03 | 0.27 – 0.29 | | |
| English channel, UK | 550 | | 0.62 – 1.04 | | | 0.01 – 72.8 | |
| Coast off Europe and French Guyana | 676 | | 0.63 – 2.07 | | 0.12 – 1.83 | 1.2 – 82.4 | Neukermans et al. (2012) |
| Guyana coast, Scheldt River, Gironde River, Rio de la Plata Estuary | 440 | 0.02 – 0.12 | | | 0.37 – 0.89 | 30 – 120 | Dogliotti et al. (2015) |

| Location | λ | | | | Reference |
|---|---|---|---|---|---|
| Elbe Estuary, Germany | 555 | 0.05 – 0.07 | 0.35 – 0.47 | 73.5 – 294.2 | Doxaran et al. (2009) |
| | 715 | 0.01 – 0.03 | 0.32 – 0.44 | | |
| Gironde Estuary, France | 555 | 0.02 – 0.06 | 0.28 – 0.50 | 21.9 – 344.1 | |
| | 715 | 0.01 – 0.02 | 0.27 – 0.45 | | |
| Coastal Louisiana and lower Atchafalaya and Mississsippi Rivers | 440 | 0.056 ± 0.012 (0.05-0.065) | | | Estapa et al. (2012) |
| | 488 | 0.035-0.05 | | | |
| | 556 | 0.25-0.35 | | | |
| | 665 | 0.125-0.02 | | | |
| West of Mississippi Delta | 443 | 0.012-0.079 | | | D'Sa et al. (2006) |
| Imperial Beach, California | 440 | 0.03-0.1 | 0.1-1.2 | 3-90 | Wozniak et al. (2010) |
| | 488 | 0.02-0.08 | 0.18-0.9 | | |
| | 556 | 0.01-0.03 | 0.2-0.9 | | |
| | 665 | 0.004-0.02 | 0.2-0.8 | | |

0.001-0.02     0.2-0.8

**Figure captions**

Figure 1. Study area. UE (green triangles), LE (blue rectangles) and SF (red circles). GSL is the Gulf of St. Lawrence.

Figure 2. Spectral variation of mass-specific optical coefficients of SPM for unfractionated samples. (a) particulate absorption at $\lambda$ = 440 nm, (b) particulate scattering at $\lambda$ = 550 nm. Each bar is the arithmetic average ± 2 standard errors as computed by using the whole dataset.

Figure 3. Spectral variation of mass-specific absorption coefficients for different size classes of suspended particulates. (a) 0.2-0.4 μm, (b)  0.4-0.7 μm, (c) 0.7-10 μm and (d) >10 μm. Curves presenting negative values at some wavelengths are not depicted. SF (black line), UE (red line) and LE (blue line).

Figure 4. Spectral variation of mass-specific scattering coefficients for different size classes of suspended particulates. (a) 0.2-0.4 μm, (b)  0.4-0.7 μm, (c) 0.7-10 μm and (d) >10 μm. Curves presenting negative values at some wavelengths are not depicted. SF (black line), UE (red line) and LE (blue line).

Figure 5. Comparison of mass-normalized optical coefficients for different SPM fractions. (a) mass-specific (left-axis) and true optical cross section (right-axis) for particulate absorption, (b) idem as (a) but for particulate scattering. Each bar is the arithmetic average ± 2 standard errors as computed by using the whole dataset.

Figure 6. Sub-regional variation of mass-specific optical coefficients of SPM. (a) particulate absorption at $\lambda$ = 440 nm, (b) particulate scattering at $\lambda$ = 550 nm. Each bar is the arithmetic average ± 2 standard errors as computed for each spatial domain.

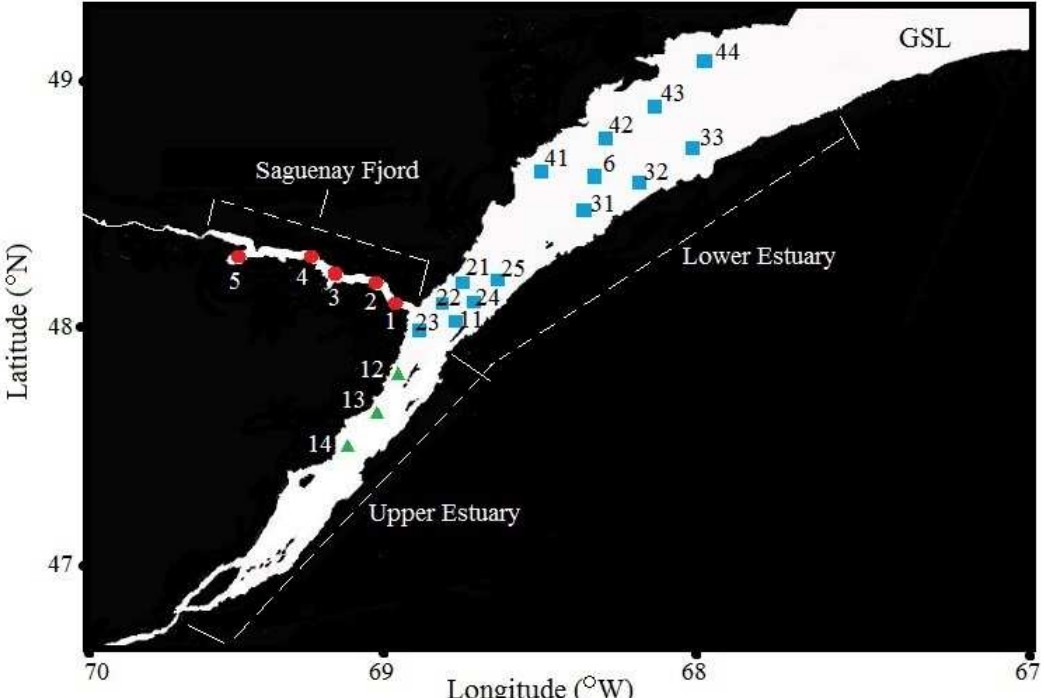

fig. 1

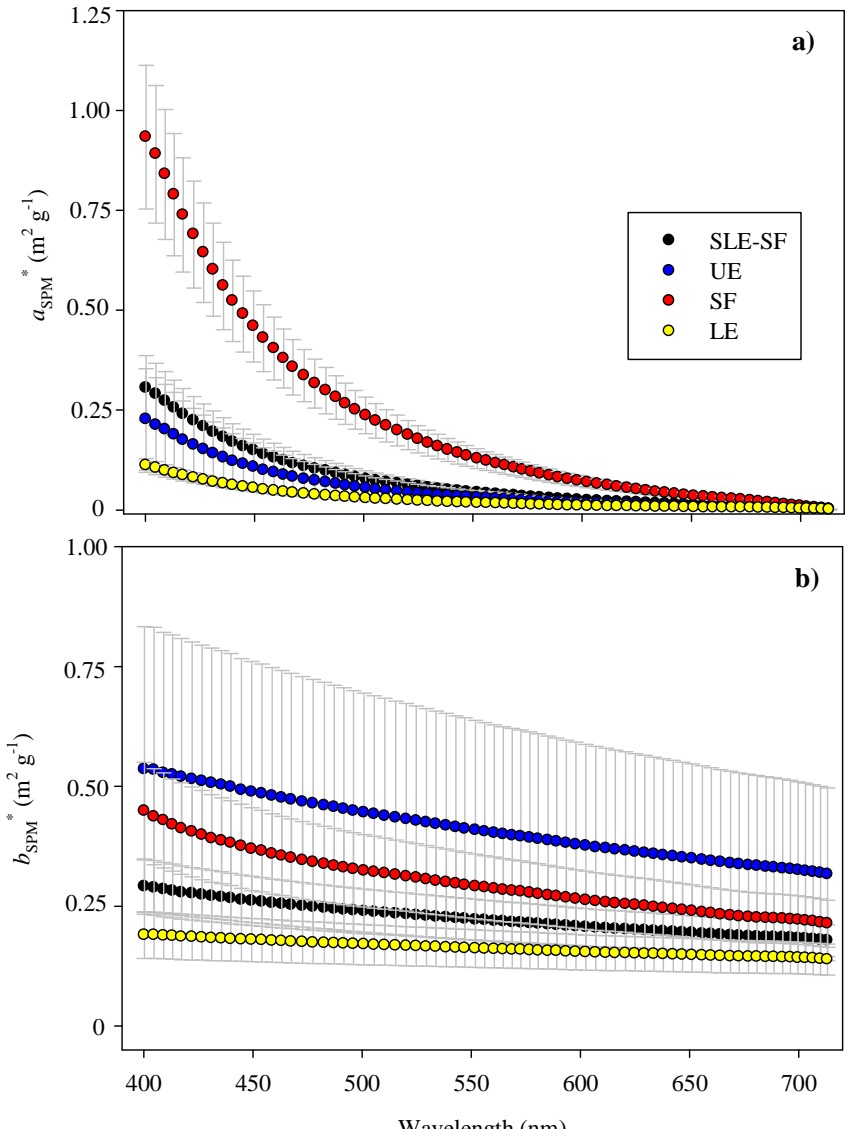

Fig. 2

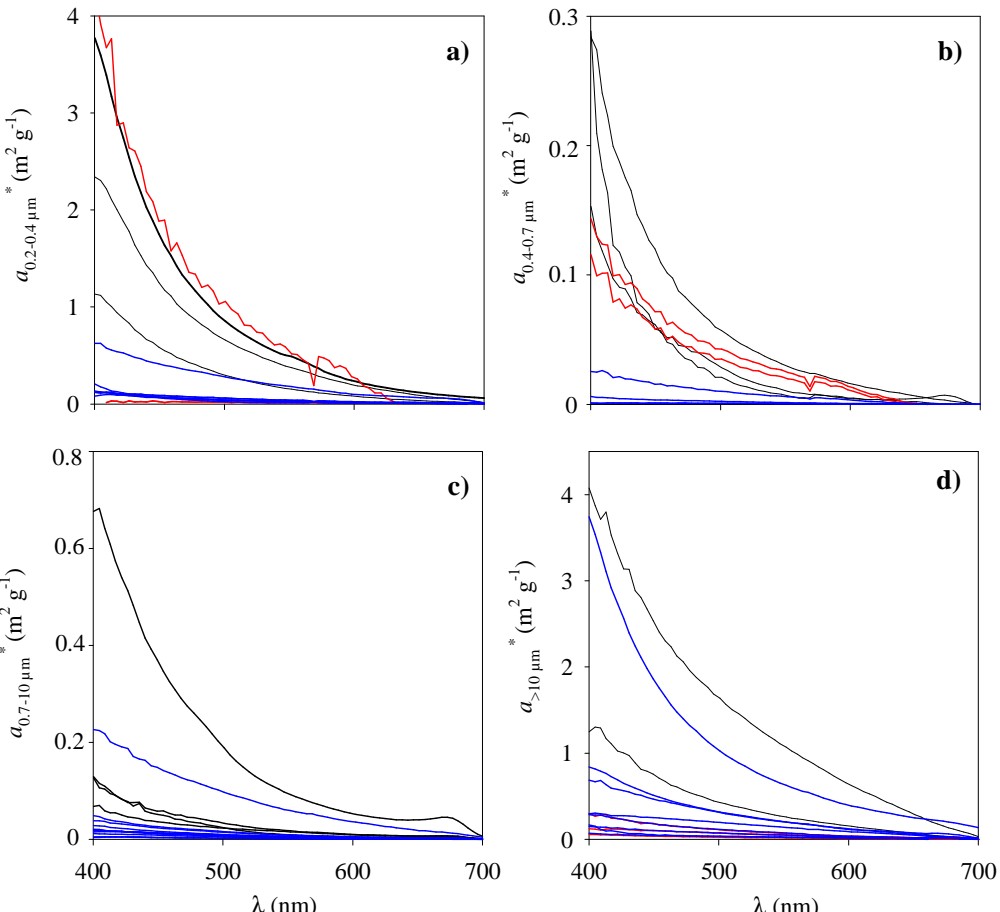

5    Fig. 3

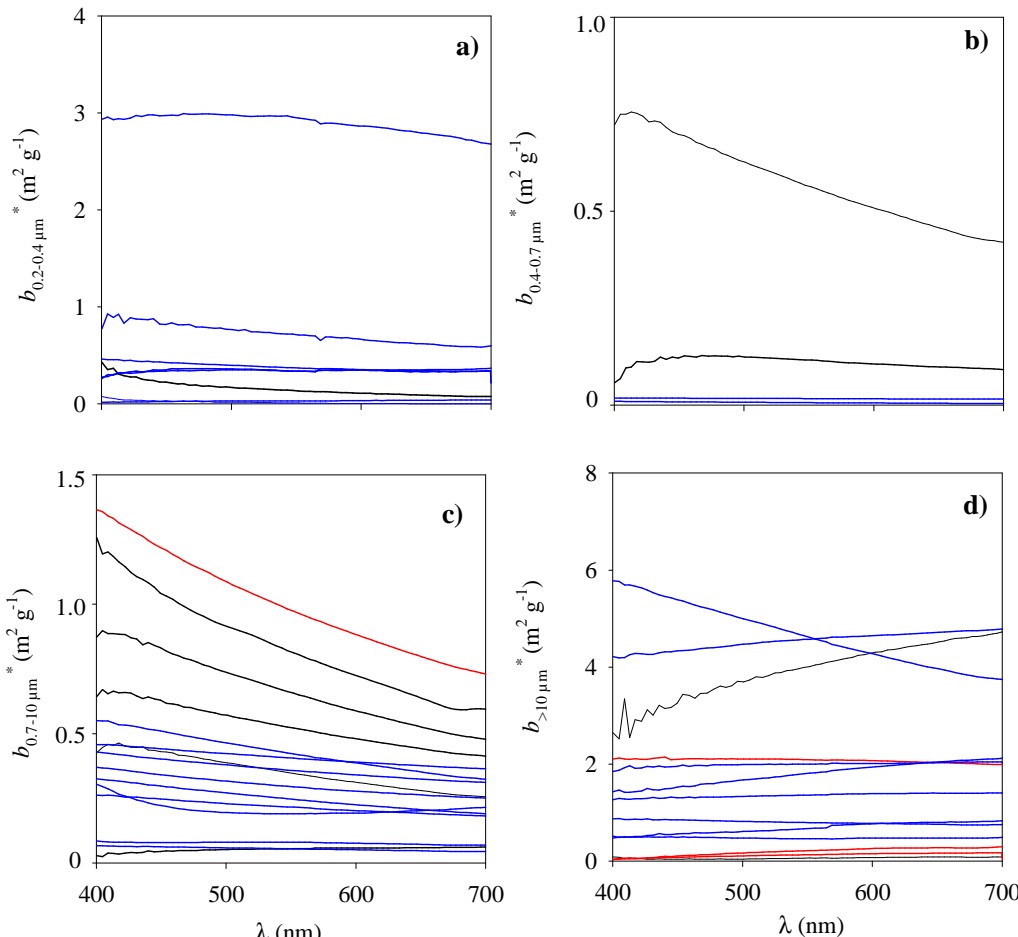

5    Fig. 4

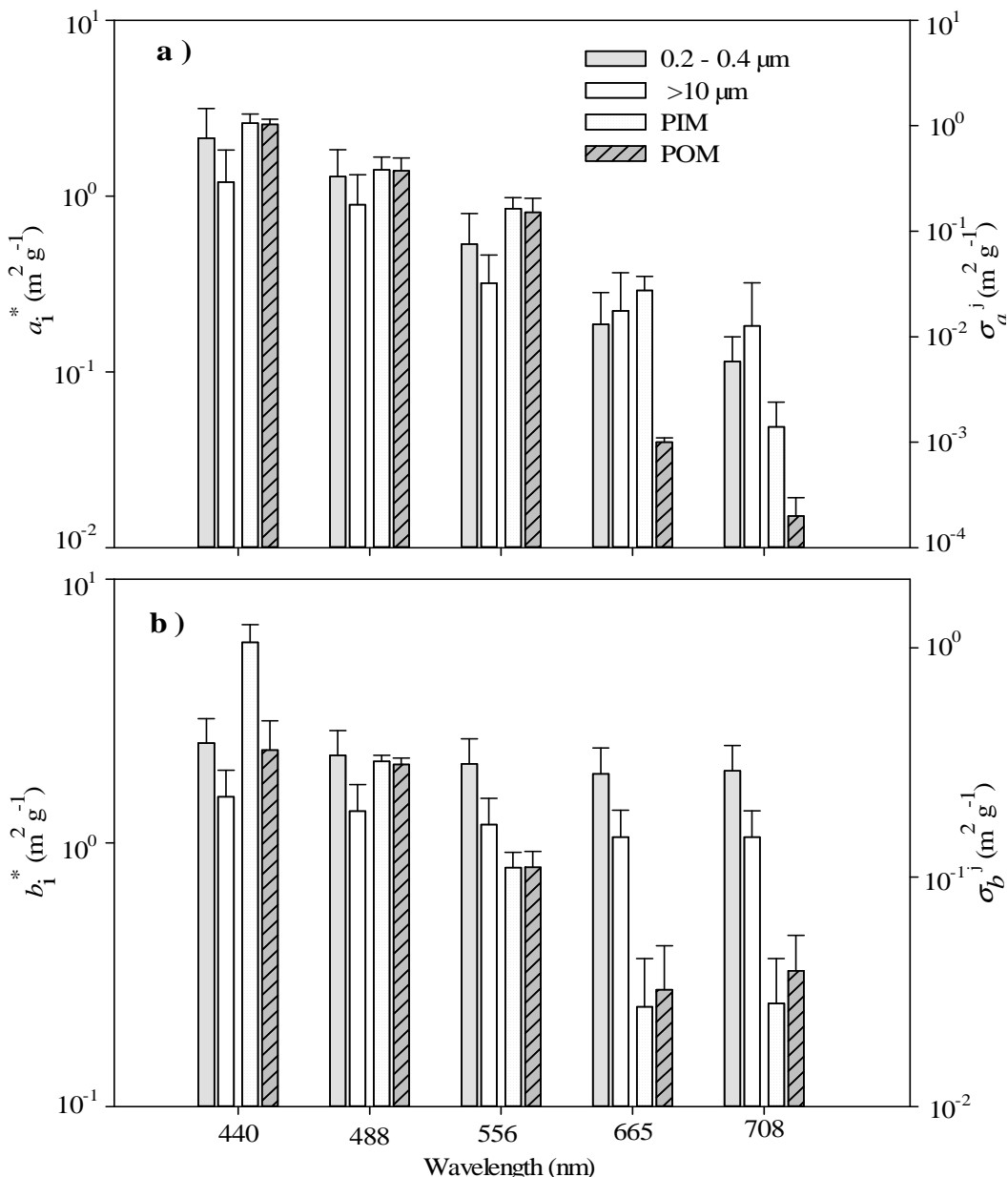

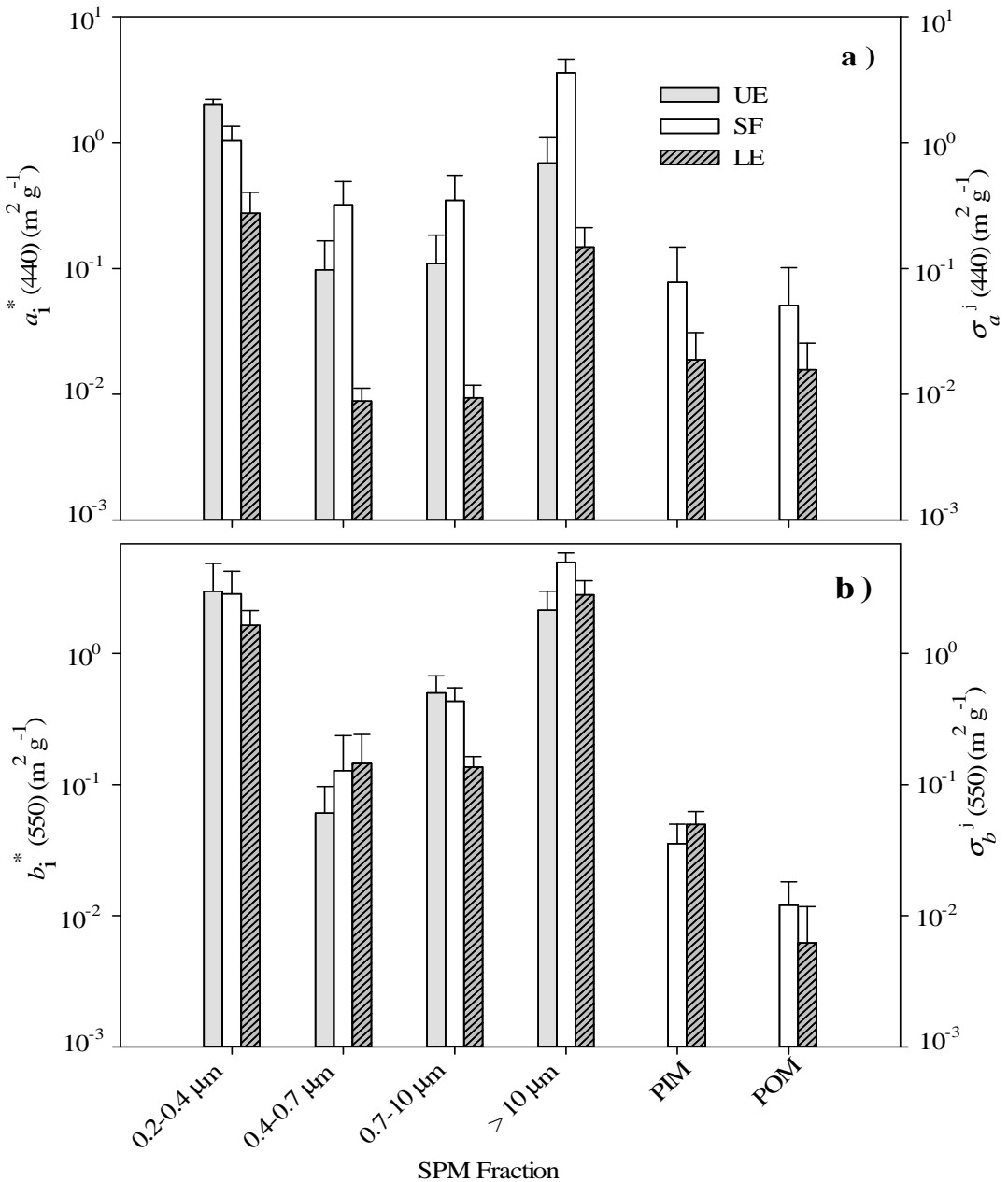

5    Fig. 6