# Peer review of "Optical properties of size and chemical fractions of suspended particulate matter in littoral waters of Quebec"

_Biogeosciences, 2017_

## Referee Comment (RC1) · Anonymous Referee #1 · 15 Jun 2017

General comment

This study looks at the variations of particulate optical properties (light absorption and scattering coefficients) as a function of the size and chemical composition (mineral/organic fractions) of the suspended particulate matter (SPM). Test sites are contrasted coastal waters: the Saint Lawrence Estuary and one adjacent fjord. Surface water samples were collected in the field and filtered into different size fractions: <0.2 m, 0.2-0.4 m, 0.4-0.7 m, 0.7-10 m and >10 m. For each size fraction, the concentration of suspended solids and optical properties (spectral (400-700) absorption and scattering coefficients) were measured. Covariations between optical proxies and biogeo-

chemical properties of SPM were examined and analysed. It is a potentially interesting experimental study which is based on a sound methodology (but several clarifications are necessary: see specific comments). The Discussion section highlights the limits of such a study due to measurement uncertainties. Unfortunately, there is no direct conclusion for ocean colour remote sensing purposes as there is no particulate backscattering measurements. The manuscript is well and clearly written. Its organization is a bit confusing with few Figures (3+1), many Tables (6+3), one Appendix.

The main comments are:

> the 'data and methods' section can be significantly improved by: (1) presenting in more details the processing of the data (notably the particle size measurements and assessment of measurement uncertainties (SPM and PIM concentrations, spectral slopes of the absorption and scattering coefficients, slope of the particle size distribution); this will highlight the quality of the dataset and give confidence to the readers (2) explaining/justifying the choice of biogeo-optical indices (BOI) (3) analysing further the relationships between the spectral slopes of the SPM optical properties (attenuation, absorption and scattering spectral slopes) and SPM size/composition (4) considering theoretical calculations (e.g., Mie theory) to support and complement the observations made on the experimental measurements (optional).

> as it, results are mainly presented as tables summarizing the observed covariations between SPM biogeochemical and optical properties, so that the study almost appears as a report of an experimental study. There is(are) no real striking result(s) highlighted in the study.

My recommendation is to improve the 'data and Methods' section partly re-organize the manuscript (no Appendix) and re-inforce the results section, notably based on my general and specific comments hereafter.

Specific comments

1. Introduction: while the objectives are clear the general methodology is not. It should be introduced and explained.

2.1 Are you the first ones measuring IOPs in this study area? If not please review past measurements.

2.2 At last you introduce here the methodology: field or lab measurements on water samples collected in the field. Why not also considering computations (e.g., Mie theory) to complement your measurements?

2.3 Uncertainties (or precisions?) on SPM and PIM concentrations (15 and 25%) seem quite high...Can you comment on this and remind your protocol? How did you estimate these 15% and 25% measurement errors?

2.4 Unclear how you measured CDOM absorption? Bench spectrophotometer? Wet-labs ac-s?

2.4 "Optical measurements were corrected by applying a flat baseline at a refer-ene wavelength of 715 nm (Bricaud and Stramski)": why using this old correction method? R. Zaneveld then Rottgers et al. (2014) have developed new correction methods...please explain

2.4 "Particle size spectra were measured by using a LISST-100X sensor": field mea-surements or measurements in the lab using the LISST as a bench sensor? Please clarify/justify. Describe how you processed the data (assuming spherical or non-spherical particles, particle size distribution, Junge exponent, etc).

2.5 Equation 1: the IOPs related to the water reflectance are the absorption and backscattering coefficients, while you measured the absorption and scattering coef-ficients. Therefore, I do not think your measurements/results can be directly used for remote sensing purposes.

2.5 Equations 3-5: can you explain/justify the choice of these biogeo-optical indices (BOI)? Notably the selection of the wavelengths?

2.5 "Values of aSPM were derived by subtracting the contributions of CDOM and sea-water to a." So aCDOM was measured using the ac-s. The ac-s sensor is calibrated in pure water, no need to subtract the contribution of seawater.

2.6 "Spectral values of mass-specific absorption and scattering cross sections for mineral and organic fractions of SPM" Please provide equations for these two parameters and physical units.

Equations 6-7: m is a concentration, not a mass

3.1 Is 'gamma' the spectral slope of particulate scattering or the Junge slope of particle size distribution? Please clarify and if possible relate these two parameters. Were Junge-type size distributions representative of the measured particle size distributions?

3.3 Mass-specific SPM absorption coefficient at 40 nm up to 4,6 m2/g: can you explain such high value? Has it been already reported in the literature?

4.1 Lines 25-30: the assumption of a negligible light absorption in the NIR (here 715 nm) is also a potential source of error (e.g., see Estapa et al. 2012, Rottgers et al. 2014). Please discuss this issue in more details.

4.1 Lines 5-10: Assuming such larges errors (>50%) on particle-related IOPs and mass-specific optical coefficients, the objective should be first to minimize these errors before analysing the results. Can you propose solutions for more accurate measurements?

4.4 'In summary, our results indicate that size (chemical composition) of suspended particulates has a major influence on spatial variability of SPM mass-specific scattering (absorption) coefficients in SLE-SF waters.' Interesting but not really a new finding.

5. Rather short conclusions as there is not many striking results presented in this study. Predicting mass-specific IOPs based on satellite remote sensing measurements is only a perspective.

---

## Referee Comment (RC2) · Anonymous Referee #2 · 29 Jun 2017

**Optical Properties of size and chemical fractions of suspended particulate matter in littoral waters of Quebec**

by

Mohammadpour, Gagné, Larouche, and Montes-Hugo

**General Comments:**

Interesting relations are reported in this paper which, however, should be used to reach better conclusions. The significance of this paper is the investigation of the partition of Suspended Particulate Matter (SPM) into its major chemical class composition of Particulate Inorganic Matter (PIM) and Particulate Organic Matter (POM) to determine true optical properties (cross sections) of the suspended matter. These true optical relations are then correlated with empirical absorption and scattering coefficients of total SPM. The true optical coefficients taken with the empirical ones allow a reasonable interpretation of the empirical coefficients of the St. Lawrence Estuary. However, the particular form taken here is to advocate a model of empirical properties of SPM that has no generality, i.e. it cannot be applied beyond the St. Lawrence Estuary. I will expand on this as I comment on specific lines and portions of the manuscript. The results of this paper document the fact that the empirical optical coefficients do not allow a general model. In sum, this paper requires a massive reanalysis of the data and a massive effort at rewriting.

**Specific Comments:**

Page 1, lines 7-9. The language in this manuscript can be pretty cryptic. Some expansion is required here and below for clarity even though I realize abstracts are supposed to be kept as short as possible.

Suggested wording: **Abstract.** Empirical mass-specific absorption ($a_{SPM}$ *) and scattering ($b_{SPM}$*) coefficients of suspended particulate matter (SPM) were measured for different size fractions (proposed to be 02.-0.4 μm, 0.4-0.7 μm, 0.7-10 μm, and >10 μm) in the surface waters (0-5 m depth) of the Saint Lawrence Estuary and Saguenay Fjords (SLE-SF) during the spring of 2013. True optical absorption and scattering cross sections were determined for the total PIM and POM, in addition to mass-specific absorption and scattering coefficients.

Page1, line 10. A synopsis of the results of the determination of the true optical absorption cross sections also needs to be reported here. It requires pulling together the results on the spectral range of absorption cross sections of at least PIM to document the effects of adsorbed iron on clay minerals or suspended iron oxides in the PIM. That is, an analysis of the true absorption cross section, $\sigma_a^j$ , for chemical fraction j, organic or inorganic. The true optical cross sections determined here provide the information to interpret the empirical coefficient ratios reported for size fractionation, etc. This information documents the statements in the final sentence of the abstract such as for the effects of chemical composition and absorption variability on what is reported here.

Page 1, lines 13-14. It is not at all clear here what the authors mean when identifying variability of the empirical mass-specific absorption and scattering coefficients. In addition, the results of this study call into question the utility and feasibility of utilizing these empirical coefficient ratios.

Suggested wording: Gironde River). $a_{SPM}$ * … particulates. Correlation analysis of the optical

properties and the empirical ratios of this study suggests that particle composition has the most significant impact on variability of $a_{SPM}$ * and particle size distribution has the most significant impact on $b_{SPM}$* variability.  The fact that knowledge of of the optical cross sections is necessary to interpret these empirical ratios calls into question the utility of $a_{SPM}$ * and $b_{SPM}$* in general models of microphysical and biogeochemical processes proposed for all coastal/estuarine systems.

A fundamental problem with all correlational analyses, as opposed to a well-defined regression analysis, is the fact that correlational analysis merely records the co-occurence of phenomena without postulating a fundamental relationship between variables of the phenomena.  High correlations simply mean that other, more fundamental relationships may be causing the co-occurence of unrelated phenomena.  An analysis based on $a_{SPM}$ * and $b_{SPM}$* will always be correlational and limited to the specific region where the relationships were defined.

Page 2, lines 1-2.  Algorithms based only on $C_{SPM}$ will never have the accuracy required for optical inversions because SPM is undefined optically, an unknown mixture of inorganic and organic matter.  Therefore partition of SPM into at least major chemical composition classes (PIM and POM) and estimation of size distribution are required independently for optically-based remote sensing algorithms of primary productivity and suspended mineral dynamics of "disappearing shorelines" etc.

Page 2, lines 13-17.  A fundamental issue here, not often discussed in the literature but should be, is consideration of what constitutes Inherent Optical Properties and how this concept should be applied to the measurements taken in the field. One can take a bulk absorption or scattering coefficient of an undefined mixture of material suspended in water and easily determine their mass-specific coefficients but what do they really mean?  The absorption coefficient of dissolved matter such as CDOM can be related to a general chemical class of dissolved compounds and we can come up with a measurement of absorption that can be related by refractive index or whatever to a similar group of compounds and the absorption coefficient of CDOM can be analyzed in a quantitative manner.  That is, an absorption coefficient of CDOM from one region can be related quantitatively (absorption  cross section, etc.) to an absorption coefficient of CDOM from an entirely different region.  So an absorption coefficient of CDOM can be called an optical property as per the definition of Bohren and Huffman (1983, p. 227), "There are two sets of quantities that are often used to describe optical properties: the real and imaginary parts of the complex refractive index $N = n + ik$ and the real and imaginary parts of the complex dielectric function (or relative permittivity) $\varepsilon = \underline{\varepsilon}' + i\varepsilon''$."  In other words, genuine optical properties must have defined complex refractive indices and permittivities which the absorption and scattering coefficients of SPM do not have. Again, SPM is an unknown mixture of both mineral and organic matter and the SPM composition varies from point to point in the same region and furthermore varies between different regions.  If we separate out mineral and organic matter we can approach true optical properties of this material by having more narrowly defined complex refractive indices and relative permittivities.  By this definition the absorption and scattering coefficients of SPM cannot be called optical properties and their mass-specific versions,  $a_{SPM}$ * and $b_{SPM}$*, should only be called empirical mass-specific ratios.  At best the absorption and scattering coefficients of unpartitioned SPM can be referred to as "optical proxies."  Thus the term "optical properties" should be limited to the optical cross sections and absorption and scattering coefficients for PIM and POM only.  This rationale will be followed in my subsequent comments.

Page 3, lines 15-20.  The description of the procedures utilized here is confusing.  The process of size-fractionation of suspended matter in water is tricky (Sheldon and Sutcliffe, 1969; Sheldon, 1972).  It is important to recognize the difference between screens and filters as was pointed out by Sheldon and Sutcliffe (1969).  A screen is designed for separation of materials in suspension of a particular diameter

and a filter is designed for retention of all materials in suspension greater than a given diameter.  That is, the manufacturer guarantees that a filter of a given nominal pore size will retain all material larger than the nominal pore size.  However, as a filter slowly gets clogged it will retain material smaller than the nominal pore size.  All the filters mentioned in this section were not designed to be screens and the nominal manufacturer's pore size is not the median pore size for retention as demonstrated by Johnson and Wangersky (1985), Sheldon (1972), and Sheldon and Sutcliffe (1969).  The median size of particles retained is a function of the volume of sample filtered and the concentration of particles in the sample.  The use of manufacturer's nominal pore size to delineate the size fractionation, as is done in this paper, does not correctly give the limits of the size fractions unless the authors did extensive tests on the particle size-range and retention capacity of the filters they utilized under their particular conditions of filtration. It is not clear which filter was used for the loss-on-ignition determination of the total suspended mass and partitioning of it into PIM and POM.  If the Whatman GF/F filter were used for SPM, PIM, and POM determination then why was the same filter used for fractionation into the supposed 0.4-0.7 and 0.7-10 μm size ranges?  The Whatman GF/F filter can work well for removing nearly all particles down to about 0.2 μm out of suspension. Chavez et al. (1995) reported about 95% particle retention down to 0.2 μm by Whatman GF/F filters.  Johnson and Wangersky (1985) derived a theory, involving diffusion and adsorption of suspended materials and filter pore walls, demonstrating that filters will retain particles much smaller than the nominal pore size reported by the manufacturer.  One of their conclusions was that a Whatman GF/C filter of nominal pore size 1.2 μm, depending on concentration of materials in suspension and flow rate of filtration, could be an efficient method of separating out materials in suspension larger than 0.7 μm

The authors need to give a table of the suspended masses retrieved by the various filter sizes and compare it with the total mass retrieved on a single filter.  Given the fact that the nominal pore sizes of the filters do not correspond to the actual sizes of material retained on the filters, I would be surprised if the masses of the sub samples from various filters added up to the total mass retrieved from one filter.  Since the masses retained by the filters are the key to the results reported in this paper, I  suggest some sort of optimization scheme to adjust the total mass and the sum of the subsample masses so that a probable mass partition can be utilized based on the masses retained on the filters, i.e. adjusting the various masses to sum up to the total mass filtered.  This would presumably require various weighting factors to be applied to the measured masses.  It appears the Whatman GF/F filters were used for both total mass filtration and for determining two sub-sample ranges.  The authors must explain carefully just how this was accomplished.

Page 3, lines 22-23. The authors point out that they did not use the correction factors discussed by Barille-Boyer et al. (2003) to account for the loss of structural water by the suspended clay minerals.  The authors state that an error of about 10% will accrue to the PIM and POM estimates if ignored.  The 10% error is only for the inorganics while the 10 % error in inorganic mass will generate a greater error in the organic mass, easily as much as 30% overestimation error in the POM estimate.  I suggest the authors utilize the extensive geochemical publications on the St. Lawrence  Estuary to estimate the probable concentration of the various clay mineral species in their samples in order to calculate this error.  One possible source is Danglejan and Smith (1973).

Page 3, line 30.  The weightings used to correct the mass fractionation of the filters should be applied here to the estimates of spectral absorption by the various estimated size fractions.

Page 4, lines 4-5.  The spectral measurements of $c$ should also be adjusted by the weightings for the SPM size fractions as mentioned above for the absorption coefficient.  Of course this then results in

weighted values of $b$ for the size fractions. Then one should check this with the known optical relation that the various $b$ values measured for the sub fractions should add up to the $b$ value recorded for the total SPM. All of this information about $a$ and $b$ values should be recorded in a table.

Page 4, lines 5-7. The authors mention the use of a LISST-100X for determining particle size spectra in the range 3-170 μm and then never mention these data again. If the data were important they should be brought into the discussion, especially considering the lack of precision and accuracy in the attempt to do size fractionation of suspended matter in this study. Were the LISST data used to estimate the Junge slope $\gamma$? If so the extensive analysis of submicron materials in this study will not have relevance to $\gamma$ and these correlations must be removed from the analysis. If the LISST-100X data were not used then the use of the LISST-100X is irrelevant to this study and should not be mentioned.

Page 4, lines 15-30. This section is totally obscure as many relationships are brought in that do not directly reflect on the studies proposed here and may have some relevance to the material at hand but I am hard put to find relationships or relevancies. The introductory material introduces the Morel and Prieur (1977) formulas for estimating Rrs that depend on measurements of backscattering $b_b$ which, however, are not used in this study. Further on, the equations (1) and (2) are supposedly used to derive Eqs. (3) – (5), the biogeo-optical (BOI) indices which do not utilize $b_b$. Clearly, the reader requires a derivation of how one gets from a backscattering formulation to a scattering formulation. The BOI indices are proposed to estimate changes in bulk chemical composition and size distribution of SPM. From these formulas of BOI indices to the end of the page the argument becomes increasingly obscure and hard to follow. It would help to write the variables used in the manuscript into fractions created by a math editor rather than the plethora of inline fractions. The inline fractions contribute to the obscurity of the argument. The relation between formulations with backscattering to derive formulations with total scattering have to be laid out clearly. The BOI for "size 1" and "size 2" have to be explained clearly. Does this refer to all the size fractionations attempted here or to just one or two? If so, which size fractions? Where does the polynomial function F come in and how do we get this from the derivation of Eqs. (4) and (5)? What is the relevance of Gordon's (1988) formulation for Rrs, in terms of $b_b$ and Eqs. (4) and (5) in terms of $b$? Again, the reader has to be led carefully from a backscattering formulation to a total scattering formulation. This section requires expansion and a total rewrite.

Page 5, line 3. Utilizing empirical relations involving POC (essentially the $CO_2$ from ashed organic matter) generalized to POM is difficult in marine systems because the crude relations between POC and POM are based on chemical analysis of detritus from higher plants. The relation between the two variables is not straightforward in marine systems because the organic content of phytoplankton differs in quantity and quality from higher plants and the various groups of phytoplankton, diatoms and prochlorophytes for example, differ from each other. I always recommend coupling POC data with POM data because of this difficulty. So relations coupling $a_{SPM}(\lambda)$, $C_{SPM}$, and POC become increasingly problematic and definitely region-specific. The derivation of the $BOI^{size}$ indices for particle size being based on the unknown spectral slope of backscattering also becomes problematic. For that matter, there is still controversy about whether there actually is a spectral slope associated with the backscattering coefficient. These indices along with $BOI^{comp}$ may be of some empirical use but they will always be regionally limited without independent information on chemical composition and size distribution to interpret them. The problem with the indices proposed here and similar indices proposed elsewhere is that they are qualitative in nature. At best, ignoring all the problems, one can only come up with qualitative "greater than or lesser than" estimates of size or chemical composition without any quantitative information which is what is needed for valid and accurate predictions of particle and biogeochemical dynamics.

Page 5, line 9.  Since the empirical mass normalizations reported here do not fit the Bohren and Huffman definition of optical properties, I suggest the following, **2.6 Optical cross sections and mass-normalized coefficients,** and the substitution of "optical coefficients or mass-specific ratios" throughout the manuscript when the term "IOP" is used to refer to the empirical mass-normalized coefficients or absorption and scattering coefficients determined for SPM.

Page 5, lines 10-11.  The mass-specific absorption and scattering cross sections were estimated with Model I multiple regression.  Just as Model II regression was used elsewhere in this paper, Model II multiple regression must be used for the best estimates of the mass-specific cross sections ($\sigma$).  The only time that Model I regressions can be used in place of Model II regressions is with a high $R^2$ value between the proposed dependent and independent variables, say $R^2 > 0.95$.  That is not the case here. It is my experience that the best estimate of slopes (as used to estimate $\sigma$ values) with $R^2$ values as low as reported here is definitely with Model II multiple regression (Stavn and Richter, 2008; Richter and Stavn, 2014).

Page 5, line 12.  In light of above I suggest in this line the replacement of "optical property" with "optical coefficient."

Page 5, line 16.  As suggested earlier, the masses used for calculating the mass-specific absorption and scattering coefficients of size-fractionated SPM should be optimized and weighted for these calculations.

Page 5, lines 23-25.  Although the slope of the power-law formulation is often used to describe the particle size distribution (PSD), the actual PSD's for estuarine systems as reported previously (Risović, 2002; Zhang et al. 2014; Zhang et al. 2017) should be mentioned.  The power-law distribution is a first-order approximation of the PSD for particles greater than about 2 μm diameter.  Therefore it will often work for total scattering with calculations involving only particles in the 2 μm + range.  It fails for sub-micron particles and since assertions are made for size fractions less than 2 μm diameter, the use of this assumption becomes questionable.  Again, the use of the γ slope gives a qualitative feeling for a relative distribution of large and small particles in suspension but fails when quantitative relations are desired. This is especially true of backscattering estimates as demonstrated by Risović (2002) and there seem to be analogies being made for parameters proposed and used in this paper that are based on backscattering.  The bottom line again is a development here that is qualitative at best and regionally limited.

Page 5, lines 25-27.  It is important to keep in mind here that the SPM parameters proposed and the SPM relations utilized in this paper are only useful when correlated with actual determinations of chemical species and some independent estimation of the size-classes of PSD.  The authors claim that functionalities between "IOP's" and BOI indices were investigated with linear regression analyses. However, I see no report of regression coefficients in the data tables, only correlation coefficients.

Page 6, lines 7-11.  How were the γ slope's calculated?  Were they from the masses of the various size sub-ranges or from the LISST data?  This is important because of the rampant inconsistencies between the size fraction masses and the γ slope estimates.  The mode of calculation must be delineated and the data shown in a table to be able to evaluate what is reported here.  Even though the largest mass of 0,2-0.4 mm particles is reported for the LE the smallest γ slope is reported for this region.  The 90% error for the γ coefficient, for which the area is not delineated here, is strong evidence for the inability of the

Junge-type slope to describe, even qualitatively, the PSD patterns for this study. A table is required for the $\gamma$ slopes and their errors.

Page 8, lines 7-9. Even if the n value is low for correlations between the BOI indices and the optical cross sections, this is the only way to validate the BOI indices and the correlations should be given with proper caveats.

Page 8, line 10. The Discussion section, in general, reads too much like a Results section. We should assume that the relevant statistical relations are in the results section and here we are interested only in the overall pattern of the results and the explanation of the patterns laid out in the results section.

Page 9, lines 22-23. The assertion here is that a larger portion of large particles and lower $\gamma$ slope's (how were they calculated?) were found in the LE region. Yet Table 2 indicates that parts of the LE region had the greatest contribution of 0.2-0.4 µm particles and a contribution of particles greater than 10 µm equivalent to or less than that of SF and UE. Here is an obvious problem with the $\gamma$ slope. The interpretation that these large particles may have been organic in nature contributes to the uncertainty of the interpretation of the SPM-based coefficients and measurements advocated in this paper.

Page 10, lines 16-17. Table 4 is nearly incomprehensible. The extensions of the table without the columns being identified is what makes the table incomprehensible. And again, in this section the statistical tests should be in the results section and we are interested only in the interpretations of the patterns in the results. The correlation coefficients reported in the table are low even though presumably significant. Again, if $\gamma$ were determined from the LISST data then any analysis of submicron particles and $\gamma$ is simply invalid.

Page 10, lines 24-25. The theoretical calculations of Babin et al. (2003) assumed the Junge slope $\gamma$ when estimating particle concentrations and calculating the Mie scattering based on the particle concentrations predicted by the Junge slope. Since the PSD has been demonstrated to not be Jungian ( Risović, 2002; Zhang et al. 2014; Zhang et al. 2017), especially in coastal waters, the Babin et al. (2003) results are not relevant here. Table 4 does not show any relationship between $b_{SPM}*$ and $\gamma$, i.e. $\gamma$ is not in the table at all. This closest approach is apparently in Table A1. Here we see that the correlations of the optical coefficients of nominal size fractions of SPM and $\gamma$ vary all over the map, from positive to negative, significant and non-significant, not at all supporting the hypothesis of $\gamma$ being a significant and explanatory variable in the this analysis. This also falsifies the hypothesis that absorption coefficients and ratios are parameters of use in general models of the occurrence and dynamics of suspended matter.

Page 11, lines 2-4. Again, the empirical indices proposed in this paper are poorly described and defined. What do the superscripts "size 1" and "size 2" mean? The BOI indices may be of some utility but again, like all similar indices based on empirical coefficients of total SPM, they are strictly qualitative in nature. The unknowns in the bulk coefficients in their definition will always cast doubt on their interpretation if ancillary evidence on PSD and composition are not available.

Page 11, lines 20-21. Suggested wording: These relationships will be useful in investigating local and regionally-limited relationships and properties of SPM. Without separate independent studies of true optical properties of PIM and POM, and of PSD, these relationships will remain problematical.

**Technical Corrections:**

Page 2, line 5.  Bowers et al. (2009) reported estimates of mass-specific scattering coefficients of and biogeo-physical characteristics of PIM, not SPM.

Page 6, line 26.  Replace "properties" with "coefficients."

Page 9, lines 9-10.  The English usage here is nearly incomprehensible.  Correct this and similar constructions with a native speaker of English.

Pages 19-30.  The tables presented here are nearly impossible to interpret.  The table extensions to multiple pages have incomplete columns and no captions to the columns.  The the table captions are limited and cryptic.

**References:**

Bohren, C.H. and D.R. Huffman. 1983. Absorption and Scattering of Light by Small Particles. John Wiley and sons, New York. *Xiv* + 530 pp.

Chavez, F.P., Buck, K.R., Bidigare, R.R., Karl, D.M., Hebel, D., Latasa, M., and Campbell, L. 1995. On the chlorophyll a retention properties of glass-fiber GF/F filters. Limnol. Oceanogr., 40: 428–433.

Johnson, B. D. and P. J. Wangersky. 1985. Seawater filtration: Particle flow and impaction considerations. Limnol. Oceanogr. 30: 966-971, doi:10.4319/lo.1985.30.5.0966.

Richter, S.J. and R.H. Stavn. 2014. Determining functional relations in multivariate oceanographic systems: Model II multiple linear regression. Journal of Atmospheric and Oceanic Technology, 31: 1663-1672.

Risović, D. 2002. Effect of suspended particulate-size distribution on the backscattering ratio in the remote sensing of seawater.  Appl. Opt., 41(33): 7092-7101.

Sheldon, R.W. 1972. Size separation of marine seston by membrane and and glass-fiber filters. Limnol. Oceanogr.,17(6): 494-498.

Sheldon, R.W. and W.H. Sutcliffe, Jr. 1969. Retention of marine particles by screens and filters. Limnol. Oceanogr., 14(3): 441-444.

Zhang, X., Stavn, R.H., Falster, A.U., Rick, J.J., Gray, D. and R.W. Gould, Jr. 2017. Size distributions of coastal ocean suspended particulate inorganic matter:Amorphous silica and clay minerals and their dynamics. Estuarine, Coastal and Shelf Science. DOI: 10.1016/j.ecss.2017.03.025

---

## Author Comment (AC1) · 25 Jul 2017

Page 1 line 14. 'high concentrations of chromophoric dissolved organic matter', why this matters? Originally was suggested since CDOM may create a coating around the particle that is expected to enhance the light absorption per unit of weight. However, this effect is likely minor, thus this CDOM influence was deleted from the text

Page 3 line 20. How PIM and POM were estimated? We expand the sentence as: 'The inorganic fraction of SPM (i.e., particulate inorganic matter or PIM) was obtained after removing the organic fraction (i.e., particulate organic matter or POM) of the original sample by combustion at 450°C for 6 h. Due to the dehydration of clays, this procedure

may introduce an additional uncertainty of -10% and +10% on particulate inorganic (PIM) and organic matter (POM), respectively (Barillé-Boyer et al., 2003; Stavn et al., 2009)'.

Page 3 line 26, CDOM is the fraction below 0.2um typically. Here you seem to call any fraction passing a filter CDOM. Make sure people understand that or use a different name, for example filtered fractions.

We agree and clarify: 'CDOM is defined here as the fraction or dissolved organic matter passing trough a membrane with a nominal pore size of 0.2 ïA■m'.

Page 3 line 30, I believe this type of method has been used way before Rottgers. Yes, the 'flat method' it was originally proposed by Bricaud and Stramski (1990) The reference was added

What did you hope to achieve with a baseline correction (e.g. Zaneveld et al., )? This is a first order correction for scattering effects on non-water absorption coefficient estimates. Is it reasonable to assume scattering is spectrally flat? It is an assumption and there is debate. Some studies have reported a spectral dependency on volume scattering functions or particulate backscattering ratios (Chami et al., 2006; McKnee et al., 2009;McKnee et al., 2013). But there is doubt regarding if this assumption can be generalized (McKnee et al., 2013).

Why did you choose this scattering correction (e.g. as opposed to the proportional or Rottgers one)? We are aware that other methods exist (e.g., proportional to wavelength, Monte Carlo) (Zaneveld et al., 1994; McKnee et al., 2013). However, the performance of these techniques to correct for residual scattering is not satisfactory either and/or may require additional optical information that we didn't have during the field surveys (e.g., particulate backscattering ratio)( McKnee et al., 2013).

Also we say, 'Thus, the calculation of particulate absorption coefficients is expected to have a bias with respect to true values measured using absorption-meter instruments

that are less influenced by particulate scattering (e.g., point-source integrating-cavity absorption meters) (Röttgers et al., 2014).

Page 4 line 20. It looks like you are using something similar to Gordon's formulation (replacing bb with b). Why not do it from the get go and skip equations (1) and (2)?

These optical proxies were deleted and no need to make reference to Gordon's formulation. We work now with two optical proxies commonly used in the literature, ïĂăïĄğ and Svis.

Page 4 line 25, aSPM(ïĄň6)/aSPM(ïĄň4) This is basically an indicator for [Chl]. You will get a better one by doing a line-height subtraction.

BOI indexes are not longer part of the manuscript

Page 4 Line 26. you may want to look Boss et al., 2004, JGR & 2009, LOM, for the use of a(676)-line height/c(660) for particulate composition.

Thanks for the advice. We checked two indices of particle composition suggested in Boss et al. (2004) JGR and Boss et al. (2009) LOM. The first index relates bb/c to POC/SPM and the second index relates bbp/bp to chlorophyll concentration/cp. Although very interesting, these two proxies were not evaluated since no backscattering measurements were obtained during our surveys.

Page 4 Line 29. Don't forget you have the spectral-slope of beam attenuation to work with as well Yes, we are aware of relationships between the spectral slope of cp and the hyperbolic slope of the particle size distribution (Boss et al., 2001). Additional correlations of cp spectral slope values were included as part of the analysis

Page 5 line 1, 'it will be very useful for an optical oceanographer evaluating your result if they could see figures of the SPECTRA of the mass normalized IOPs.

We added one additional figure (fig 2) where averaged ap* and bp* for the whole study area and each subregion are shown as a function of wavelength are shown

Page 5 Line 5, 'why not also compute c* for which there is a longer literature?' Although it is possible and interesting, our main interest is focused in IOPs that separate scattering from absorption effects. This is not the case for c or cp, thus their use makes interpretation of optical processes less clear.

Page 5 line 12, 'you may want to refer to it as the exponent of the power-law distribution. Junge is usually used to denote the one with a differential exponent of 4' Done

Page 5 line 16, 'you can use D50 from the LISST as a more robust parameter' Additional correlations between parameter D50 (here symbolized with Dm), spectral slope of particulate beam attenuation (ïAğ), differential slope particle size distribution (ïAÿ), mass fraction of PIM or concentration of PIM/concentration of SPM ratio (FSPMPIM), and mass-specific optical coefficients did not show a general improvement with respect to parameter ïAğ (see below). The correlation will depend on the size fraction

Correlations in the following table are based on 23 sampling locations. FSPMPIM Dm ïAğ ïAÿ FSPM0.2 – 0.4 $\mu$m -0.42* -0.51* 0.53* -0.28 ** FSPM0.4 – 0.7 $\mu$m 0.35 0.41 * -0.43* 0.11 * FSPM0.7 – 10 $\mu$m 0.23 0.08 -0.38* 0.12 * FSPM>10 $\mu$m -0.08 0.21 0.13 -0.04

Page 7 line 1, 'you paper is totally lacking an uncertainty analysis. You need to add uncertainties in all your calculated values based on:replication. Assumptions (e.g. scattering correction used, finite acceptance angle of the ac-9)'.

One additional section 4.1 was included in discussion to summarize the different uncertainities involved in measuring IOPs. There was not replication of discrete samples, however it was possible to compute the optical variability during the ac-s measurements. This information is described in discussion along with the assumptions regarding the trasmissometer and the scattering effects on a estimates.

Page 7 line 16, 'not having backscattering measurements and radiometry, this is a hard case to make.' We agree , we talk now about optical proxies instead of remote sensing

proxies

Page 8 line 22, 'But note that multiple scattering may have affected their optical measurements' We added this observation to the discussion. 'Notice that part of this decrease can be attributed to an incomplete removal of multiple scattering effects'.

Page 8 line 24, 'This is known for a long while, e.g. Morel's 1974 work' We added this reference

Page 8 line 31, 'CDOM cannot explain increase in a_SPM*' The effect of CDOM on ap* was not quantified and is likely to be minor, thus it is a weak statement. Thus it was deleted from the text

Page 9 line 12, 'this will be true for in-situ aggregates (Slade's work). However, you are disrupting aggregates, so it is less likely'

We clarify the sentence as follows: 'Since particle aggregates were altered during our experiments, the influence of particle density on mass-specific optical coefficients cannot be quantified as this effect is mainly observed in undisrupted marine aggregates (Slade et al. 2011..'

---

## Author Comment (AC2) · 25 Jul 2017

General comments In sum, this paper requires a massive reanalysis of the data and a massive effort at rewriting.

The corrected version of the manuscript was totally re-structured, many sections rewritten, new figures, new analysis of data and deletion of redundant tables

Specific Comments: Page 1, lines 7-9. The language in this manuscript can be pretty cryptic. Some expansion is required here and below for clarity even though I realize abstracts are supposed to be kept as short as possible.Suggested wording: Abstract.

[Figure]

Empirical mass-specific absorption (aSPM *) and scattering (bSPM*) coefficients of suspended particulate matter (SPM) were measured for different size fractions (proposed to be 02.-0.4 ïÅăïĄ∎m, 0.4-0.7 ïĄ∎m, 0.7-10 ïĄ∎m, and >10 ïĄ∎m) in the surface waters (0-5 m depth) of the Saint Lawrence Estuary and Saguenay Fjords (SLE-SF) during the spring of 2013. True optical absorption and scattering cross sections were determined for the total PIM and POM, in addition to mass-specific absorption and scattering coefficients.

The abstract was rewritten and we talk now about true optical absoprtiona nd scattering cross sections

Page1, line 10. A synopsis of the results of the determination of the true optical absorption cross sections also needs to be reported here. It requires pulling together the results on the spectral range of absorption cross sections of at least PIM to document the effects of adsorbed iron on clay minerals or suspended iron oxides in the PIM. That is, an analysis of the true absorption cross section, aj , for chemical fraction j, organic or inorganic. The true optical cross sections determined here provide the information to interpret the empirical coefficient ratios reported for size fractionation, etc. This information documents the statements in the final sentence of the abstract such as for the effects of chemical composition and absorption variability on what is reported here.

Spectral changes on aspm* are now reported and discussed in terms of iron effects on particulate absorption.

Page 1, lines 13-14. It is not at all clear here what the authors mean when identifying variability of the empirical mass-specific absorption and scattering coefficients. In addition, the results of this study call into question the utility and feasibility of utilizing these empirical coefficient ratios. Suggested wording: Gironde River). aSPM * . . . particulates. Correlation analysis of the optical properties and the empirical ratios of this study suggests that particle composition has the most significant impact on variability of aSPM * and particle size distribution has the most significant impact

on bSPM* variability. The fact that knowledge of of the optical cross sections is necessary to interpret these empirical ratios calls into question the utility of aSPM * and bSPM* in general models of microphysical and biogeochemical processes proposed for all coastal/estuarine systems.

The abstract was rewritten and results clarified

A fundamental problem with all correlational analyses, as opposed to a well-defined regression analysis, is the fact that correlational analysis merely records the cooccurence of phenomena without postulating a fundamental relationship between variables of the phenomena. High correlations simply mean that other, more fundamental relationships may be causing the co-occurence of unrelated phenomena. An analysis based on aSPM * and bSPM* will always be correlational and limited to thespecific region where the relationships were defined.

Like any other statistical analysis there are pros and cons. A weel-defined regression analysis has many and strict assumptions that should be met such as normal distribution of variables, random sampling, etc. This issue is absent when non-parametric correlations are used Yes, we are always talking about our study area regarding correlations results

Page 2, lines 1-2. Algorithms based only on CSPM will never have the accuracy required for optical inversions because SPM is undefined optically, an unknown mixture of inorganic and organic matter. Therefore partition of SPM into at least major chemical composition classes (PIM and POM) and estimation of size distribution are required independently for optically-based remote sensing algorithmsof primary productivity and suspended mineral dynamics of "disappearing shorelines" etc.

It is a relative questioning. Depends on the level of accuracy you are interested. Many studies have been proposed for estimating CSPM based on remote sensing methods. Adding remotely sensed PIM and POM to calculate SPM will also have a large error due to the addition of two errors linked to PIM and POM algorithms

Page 2, lines 13-17. A fundamental issue here, not often discussed in the literature but should be, is consideration of what constitutes Inherent Optical Properties and how this concept should be applied to the measurements taken in the field. One can take a bulk absorption or scattering coefficient of an undefined mixture of material suspended in water and easily determine their mass-specific coefficients but what do they really mean? The absorption coefficient of dissolved matter such as CDOM can be related to a general chemical class of dissolved compounds and we can come up with a measurement of absorption that can be related by refractive index or whatever to a similar group of compounds and the absorption coefficient of CDOM can be analyzed in a quantitative manner. That is, an absorption coefficient of CDOM from one region can be related quantitatively (absorption cross section, etc.) to an absorption coefficient of CDOM from an entirely different region. So an absorption coefficient of CDOM can be called an optical property as per the definition of Bohren and Huffman (1983, p. 227), "There are two sets of quantities that are often used to describe optical properties: the real and imaginary parts of the complex refractive index $N = n + ik$ and the real and imaginary parts of the complex dielectric function (or relative permittivity) $e = e' + ie$"." In other words, genuine optical properties must have defined complex refractive indices and permittivities which the absorption and scattering coefficients of SPM do not have. Again, SPM is an unknown mixture of both mineral and organic matter and the SPM composition varies from point to point in the same region and furthermore varies between different regions. If we separate out mineral and organic matter we can approach true optical properties of this material by having more narrowly defined complex refractive indices and relative permittivities. By this definition the absorption and scattering coefficients of SPM cannot be called optical properties and their mass-specific versions, aSPM * and bSPM*, should only be called empirical mass-specific ratios. At best the absorption and scattering coefficients of unpartitioned SPM can be referred to as "optical proxies." Thus the term "optical properties" should be limited to the optical cross sections and absorption and scattering coefficients for PIM and POM only. This rationale will be followed in my subsequent comments.

I understand your point and thank you for your wonderful insight but we try to go along with definitions of optical properties currently used in the current literature. This should be a good topic for another publication.

Page 3, lines 15-20. The description of the procedures utilized here is confusing. The process of sizefractionation of suspended matter in water is tricky (Sheldon and Sutcliffe, 1969; Sheldon, 1972). It is important to recognize the difference between screens and filters as was pointed out by Sheldon and Sutcliffe (1969). A screen is designed for separation of materials in suspension of a particular diameter and a filter is designed for retention of all materials in suspension greater than a given diameter. That is, the manufacturer guarantees that a filter of a given nominal pore size will retain all material larger than the nominal pore size. However, as a filter slowly gets clogged it will retain material smaller than the nominal pore size. All the filters mentioned in this section were not designed to be screens and the nominal manufacturer's pore size is not the median pore size for retention as demonstrated by Johnson and Wangersky (1985), Sheldon (1972), and Sheldon and Sutcliffe (1969). The median size of particles retained is a function of the volume of sample filtered and the concentration of particles in the sample. The use of manufacturer's nominal pore size to delineate the size fractionation, as is done in this paper, does not correctly give the limits of the size fractions unless the authors did extensive tests on the particle size-range and retention capacity of the filters they utilized under their particular conditions of filtration. It is not clear which filter was used for the loss-on-ignition determination of the total suspended mass and partitioning of it into PIM and POM. If the Whatman GF/F filter were used for SPM, PIM, and POM determination then why was the same filter used for fractionation into the supposed 0.4-0.7 and 0.7-10 mm size ranges? The Whatman GF/F filter can work well for removing nearly all particles down to about 0.2 mm out of suspension. Chavez et al. (1995) reported about 95% particle retention down to 0.2 mm by Whatman GF/F filters. Johnson and Wangersky (1985) derived a theory, involving diffusion and adsorption of suspended materials and filter pore walls, demonstrating that filters will retain particles much smaller than the nominal pore size reported by the manufacturer.One of their conclusions was that a Whatman GF/C filter of nominal pore size 1.2 mm, depending on concentration of materials in suspension and flow rate of filtration, could be an efficient method of separating out materials in suspension larger than 0.7 mm

No, different filters were used in this study to fractionate 0.4-0.7 and 0.7-10 micron fractions This sentence was clarified as:

Size fractionation of SPM into four size classes (>10 $\mu$m, 0.7-10 $\mu$m, 0.4-0.7 $\mu$m, and 0.2-0.4 $\mu$m) was done after sequentially filtering the original samples through pre-weighted membranes having a diameter of 47 mm and a pore size of 10 $\mu$m (Whatman, polycarbonate), 0.7 $\mu$m (GF/F, Whatman, glass fiber), 0.4 $\mu$m (Whatman, polycarbonate), and 0.2 $\mu$m (Nucleopore, polycarbonate), respectively.

Also, we add the whatman GF/F was used for PIM and POM determinations.

We wrote: The mass of PIM was obtained after removing the organic fraction (i.e., POM) from the total mass of SPM as computed for CSPM determinations. The mass of POM was eliminated by combustion of GF/F filters at 450°C and during 6 h. The concentration of POM was calculated as the difference between the dry mass of SPM and the dry mass of PIM. The precision of PIM determinations was 25% since an additional variability of 10% was added to the error measurement of SPM mass due to the dehydration

We are aware of the issues using the nominal pore size and the retention of small particulates. We mention that issue when computing mass-specific optical coefficients in discussion

The authors need to give a table of the suspended masses retrieved by the various filter sizes and compare it with the total mass retrieved on a single filter. Given the fact that the nominal pore sizes of the filters do not correspond to the actual sizes of material retained on the filters, I would be surprised if the masses of the sub samples from

various filters added up to the total mass retrieved from one filter. Since the masses retained by the filters are the key to the results reported in this paper, I suggest some sort of optimization scheme to adjust the total mass and the sum of the subsample masses so that a probable mass partition can be utilized based on the masses retained on the filters, i.e. adjusting the various masses to sum up to the total mass filtered. This would presumably require various weighting factors to be applied to the measured masses. It appears the Whatman GF/F filters were used for both total mass filtration and for determining two sub-sample ranges. The authors must explain carefully just how this was accomplished.

Good point. Total mass of SPM was calculated based on gravimetric determinations based on 0.7 microns GF/F filters. This is standard in the literature (Stavn and Richter, 2008; Rottger et al, 2014). However, it is true that aspm* and bspm* are overestimated since aspm and bspm are based on particulates above 0.2 microns This is due to the pre-filtration of samples through nucleopore membranes in order to remove CDOM+seawater contributions.

In the other hand, size fractions of IOPs correspond to the same size fractions of mass. Thus, there should be no bias on mass-specific coefficients of IOPs for SPM. To evaluate the effect of sieving on retaining smaller particulates than pore size, comparison were made between filtered samples without pre-sieving vs sum of size-fractioned samples. In average, adding mass fractions resulted in a total mass difference for particulates larger 0.7 microns of +31.4%. In other words, a 31.4% overestimation of mass for >0.7 microns particulates when the sum of weights of fractions is performed rather than weighting the unfractionated sample

These filter 'effects' on retained SPM mass may be possible to correct as the example described above. However, we didn't filter total unfiltered samples through 0.2 or 0.4 microns membranes since we did sequential filtering. Thus, factors such as sum fractions mass/unfiltered mass for size fractions >0.2 or >0.4 microns could not be calculated in our study.

In discussion, we described the general overestimation of aspm* and bspm* values and 'filter effects' on mass-specific properties of size fractions of SPM

Page 3, lines 22-23. The authors point out that they did not use the correction factors discussed by Barille-Boyer et al. (2003) to account for the loss of structural water by the suspended clay minerals. The authors state that an error of about 10% will accrue to the PIM and POM estimates if ignored. The 10% error is only for the inorganics while the 10 % error in inorganic mass will generate a greater error in the organic mass, easily as much as 30% overestimation error in the POM estimate. I suggest the authors utilize the extensive geochemical publications on the St. Lawrence Estuary to estimate the probable concentration of the various clay mineral species in their samples in order to calculate thiserror. One possible source is Danglejan and Smith (1973).

We added to the text the larger error of POM mass determinations By using the Danglejan and Smith (1973) data related to clay composition in the SLE, we calculated an underestimation of PIM mass of 3.1% Also, 3.22% of loss of ignition PIM must be removed from POM in order to obtain a POM mass corrected by structural water of clays

Based on Barillé-Boyer et al. (2003) factors and clay composition data obtained in the Saint Lawrence Estuary (D'Anglejan and Smith, 1973), the estimated error of PIM determinations due to dehydration of clays was 3.1%. Thus, PIM mass determinations has a maximum uncertainty of 18.1%. Notice that error in POM mass estimates is slightly greater than that associated to PIM mass (18.22% of loss on ignition PIM mass

Page 3, line 30. The weightings used to correct the mass fractionation of the filters should be applied here to the estimates of spectral absorption by the various estimated size fractions.

See above the issue of using this weighting for certain size fractions

Page 4, lines 4-5. The spectral measurements of c should also be adjusted by the

weightings for the SPM size fractions as mentioned above for the absorption coefficient. Of course this then results in weighted values of b for the size fractions. Then one should check this with the known optical relation that the various b values measured for the sub fractions should add up to the b value recorded for the total SPM. All of this information about a and b values should be recorded in a table.

Actually IOPs after each filtration are not added up but are decreasing in magnitude as samples are filtered through membranes having a smaller pore size New figures are shown for size fractions of aspm* and bspm*. For some samples, the calculation of IOPs lead to negative values at some wavelengths. These curves are not included as part of the plots and might be related to issues linked to the filters or particle aggregation/disaggregation effects

Page 4, lines 5-7. The authors mention the use of a LISST-100X for determining particle size spectra in the range 3-170 mm and then never mention these data again. If the data were important they should be brought into the discussion, especially considering the lack of precision and accuracy in the attempt to do size fractionation of suspended matter in this study. Were the LISST data used to estimate the Junge slope g? If so the extensive analysis of submicron materials in this study will not have relevance to g and these correlations must be removed from the analysis. If the LISST-100X data were not used then the use of the LISST-100X is irrelevant to this study and should not be mentioned.

Yes, LISST-100x was an important instrument to compute differential Junge slope. More interpretation and results and included now regarding ïAÿ We don't think correlations between smaller than 2 microns particulates and ïAÿ are spurious since it is feasible correlations due to the fact than smaller than 2 microns optics is correlated with greater than 2 microns optics. We verified that possibility.

Page 4, lines 15-30. This section is totally obscure as many relationships are brought in that do not directly reflect on the studies proposed here and may have some relevance

to the material at hand but I am hard put to find relationships or relevancies. The introductory material introduces the Morel and Prieur (1977) formulas for estimating Rrs that depend on measurements of backscattering bb which, however, are not used in this study. Further on, the equations (1) and (2) are supposedly used to derive Eqs. (3) – (5), the biogeo-optical (BOI) indices which do not utilize bb . Clearly, the reader requires a derivation of how one gets from a backscattering formulation to a scattering formulation. The BOI indices are proposed to estimate changes in bulk chemical composition and size distribution of SPM. From these formulas of BOI indices to the end of the page the argument becomes increasingly obscure and hard to follow. It would help to write the variables used in the manuscript into fractions created by a math editor rather than the plethora of inline fractions. The inline fractions contribute to the obscurity of the argument. The relation between formulations with backscattering to derive formulations with total scattering have to be laid out clearly. The BOI for "size 1" and "size 2" have to be explained clearly. Does this refer to all the size fractionations attempted here or to just one or two? If so, whichsize fractions? Where does the polynomial function F come in and how do we get this from the derivation of Eqs. (4) and (5)? What is the relevance of Gordon's (1988) formulation for Rrs, in terms of bb and Eqs. (4) and (5) in terms of b? Again, the reader has to be led carefully from a backscattering formulation to a total scattering formulation. This section requires expansion and a total rewrite.

All this section was rewritten and BOI indexes were removed and replaced by traditional indexes used in the literature (spectral slope of particulate beam attenuation and mass-specific particulate absorption coefficient within the visible spectrum). Part of decision of eliminating BOI indexes was the lack of bb measurements.

Page 5, line 3. Utilizing empirical relations involving POC (essentially the $CO_2$ from ashed organic matter) generalized to POM is difficult in marine systems because the crude relations between POC and POM are based on chemical analysis of detritus from higher plants. The relation between the two variables is not straightforward in marine

systems because the organic content of phytoplankton differs in quantity and quality from higher plants and the various groups of phytoplankton, diatoms and prochlorophytes for example, differ from each other. I always recommend coupling POC data with POM data because of this difficulty. So relations coupling aSPM(I), CSPM, and POC become increasingly problematic and definitely region-specific. The derivation of the BOIsize indices for particle size being based on the unknown spectral slope of backscattering also becomes problematic. For that matter, there is still controversy about whether there actually is a spectral slope associated with the backscattering coefficient. These indices along with BOIcomp may be of some empirical use but they will always be regionally limited without independent information on chemical composition and size distribution to interpret them. The problem with the indices proposed here and similar indices proposed elsewhere is that they are qualitative in nature. At best, ignoring all the problems, one can only come up with qualitative "greater than or lesser than" estimates of size or chemical composition without any quantitative information which is what is needed for valid and accurate predictions of particle and biogeochemical dynamics.

We agree with the reviewer. BOI indexes are no longer part of the mansucript

Page 5, line 9. Since the empirical mass normalizations reported here do not fit the Bohren and Huffman definition of optical properties, I suggest the following, 2.6 Optical cross sections and massnormalized coefficients, and the substitution of "optical coefficients or mass-specific ratios" throughout the manuscript when the term "IOP" is used to refer to the empirical mass-normalized coefficients or absorption and scattering coefficients determined for SPM.

Done Hopefully well understood. Replacing whenever IOP is present

Page 5, lines 10-11. The mass-specific absorption and scattering cross sections were estimated with Model I multiple regression. Just as Model II regression was used elsewhere in this paper, Model II multiple regression must be used for the best estimates

of the mass-specific cross sections (s). The only time that Model I regressions can be used in place of Model II regressions is with a high R2 value between the proposed dependent and independent variables, say R2 > 0.95. That is not the case here. It is my experience that the best estimate of slopes (as used to estimate s values) with R2 values as low as reported here is definitely with Model II multiple regression (Stavn and Richter, 2008; Richter and Stavn, 2014).

Sorry, model II was missing from the text. Now was added and it means that response and independent variables have a random error

Optical cross sections for chemical fractions of SPM were calculated based on multiple regression model II analysis (i.e., independent and response variables have random errors) (Sokal et al., 1995; Stavn and Richter, 2008): $Y = \beta1$ [CPIM] + $\beta2$ [CPOM]

Page 5, line 12. In light of above I suggest in this line the replacement of "optical property" with "optical coefficient." done

Page 5, line 16. As suggested earlier, the masses used for calculating the mass-specific absorption and scattering coefficients of size-fractionated SPM should be optimized and weighted for these calculations. As mentioned before, this is not possible for all size fractions

Page 5, lines 23-25. Although the slope of the power-law formulation is often used to describe the particle size distribution (PSD), the actual PSD's for estuarine systems as reported previously (Risović, 2002; Zhang et al. 2014; Zhang et al. 2017) should be mentioned. The power-law distribution is a firstorder approximation of the PSD for particles greater than about 2 mm diameter. Therefore it will often work for total scattering with calculations involving only particles in the 2 mm + range. It fails for submicron particles and since assertions are made for size fractions less than 2 mm diameter, the use of this assumption becomes questionable. Again, the use of the g slope gives a qualitative feeling for a relative distribution of large and small particles in suspension but fails when quantitative relations are desired. This is especially true

of backscattering estimates as demonstrated by Risović (2002) and there seem to be analogies being made for parameters proposed and used in this paper that are based on backscattering. The bottom line again is a development here that is qualitative at best and regionally limited.

I guess you mean The power-law distribution is a firstorder approximation of the PSD for particles greater than about 2 mm diameter, greater than 2 microns, right? We included in methods the limitations of using Junge slope vs Risovic

'Although particle size distribution in natural waters may not follow a Junge-type slope, its use here was justified since our main interest was to have a first-order assessment of size effects of particulates on optical coefficient's variability'. Indeed, the calculation of ïAÿ is only valid for particulates greater than 2 ïA■m. A more realistic representation of PSD is the model proposed by Risovic (2002). This parameterization mainly includes two particle populations ('large' and 'small') having different refractive index and was applied for the first time in littoral environments by Stavn and Richter (2008). Thus, relationships between ïAÿ and optical coefficients in this study are local and should not be generalized to other littoral environments.

Page 5, lines 25-27. It is important to keep in mind here that the SPM parameters proposed and the SPM relations utilized in this paper are only useful when correlated with actual determinations of chemical species and some independent estimation of the size-classes of PSD. The authors claim that functionalities between "IOP's" and BOI indices were investigated with linear regression analyses. However, I see no report of regression coefficients in the data tables, only correlation coefficients.

BOI indexes are not aprt of the manuscript anymore. We do correlations not linear regressions

Page 6, lines 7-11. How were the g slope's calculated? Were they from the masses of the various size sub-ranges or from the LISST data? This is important because of the rampant inconsistencies between the size fraction masses and the g slope estimates.

The mode of calculation must be delineated and the data shown in a table to be able to evaluate what is reported here. Even though the largest mass of 0,2- 0.4 mm particles is reported for the LE the smallest g slope is reported for this region. The 90% error for the g coefficient, for which the area is not delineated here, is strong evidence for the inability of the Junge-type slope to describe, even qualitatively, the PSD patterns for this study. A table is required for the g slopes and their errors.

Error and range of ïAÿ values was added to the manuscript. Also, a detailed calculation of ïAÿ is included

Page 8, lines 7-9. Even if the n value is low for correlations between the BOI indices and the optical cross sections, this is the only way to validate the BOI indices and the correlations should be given with proper caveats.

In Discussion we highlighted the limitations of the reduced number of samples when correlating optical proxies 'Also, the reduced number of sampling locations and the geographic variability of ïAÿ-ïAğ relationships were additional factors likely explaining the lack of a general functionality for the study area'

Page 8, line 10. The Discussion section, in general, reads too much like a Results section. We should assume that the relevant statistical relations are in the results section and here we are interested only in the overall pattern of the results and the explanation of the patterns laid out in the results section.

The discussion was improved with results regarding iron and new optical proxies Comparison of our mass-specific optical coefficients and optical cross sections with those in the literature is a common procedure of discussing results in most publications

Page 9, lines 22-23. The assertion here is that a larger portion of large particles and lower g slope's (how were they calculated?) were found in the LE region. Yet Table 2 indicates that parts of the LE region had the greatest contribution of 0.2-0.4 mm particles and a contribution of particles greater than 10 mm equivalent to or less than

that of SF and UE. Here is an obvious problem with the g slope. The interpretation that these large particles may have been organic in nature contributes to the uncertainty of the interpretation of the SPM-based coefficients and measurements advocated in this paper.

The calculation of the slope ïAÿ is described in methods. Yes, there is a lot spatial variability on ïAÿ but lower ïAÿ values were measured in LE waters. We found a general inverse correlation between CPIM/CSPM and ïAÿ (ïAšs = -0.41, P = 0.049) for the study area and suggesting that relatively large particulates have an organic origin. This relationship was intensified in LE waters (ïAšs = -0.58, P = 0.022).

Page 10, lines 16-17. Table 4 is nearly incomprehensible. The extensions of the table without the columns being identified is what makes the table incomprehensible. And again, in this section the statistical tests should be in the results section and we are interested only in the interpretations of the patterns in the results. The correlation coefficients reported in the table are low even though presumably significant. Again, if g were determined from the LISST data then any analysis of submicron particles and g is simply invalid.

The first column of the table was labeled. More statistical detailed were added to the text. Now is table 2. It is true that no correlations should be expected between ïAÿ and mass-specific optical coefficients of size fraction 0.2-04 and 04-0.7 microns cause the LISST limitations regarding submicrometric particles. However, correlations may exist due to dependencies between size fractions. In other words, aspm* of 0.2-0.4 and 0.4-0.7 microns are correlated with aspm* of 0.7-10 and >10 microns For the case of bspm*, no significant correlations were computed for 0.2-0.4 and 0.4-0.7 microns.

Page 10, lines 24-25. The theoretical calculations of Babin et al. (2003) assumed the Junge slope g when estimating particle concentrations and calculating the Mie scattering based on the particle concentrations predicted by the Junge slope. Since the PSD has been demonstrated to not be Jungian ( Risović, 2002; Zhang et al. 2014;

Zhang et al. 2017), especially in coastal waters, the Babin et al. (2003) results are not relevant here. Table 4 does not show any relationship between bSPM* and g , i.e. g is not in the table at all. This closest approach is apparently in Table A1. Here we see that the correlations of the optical coefficients of nominal size fractions of SPM and g vary all over the map, from positive to negative, significant and non-significant, not at all supporting the hypothesis of g being a significant and explanatory variable in the this analysis. This also falsifies the hypothesis that absorption coefficients and ratios are parameters of use in general models of the occurrence and dynamics of suspended matter.

The sentence about Babin et al. (2003) was deleted in discussion

Page 11, lines 2-4. Again, the empirical indices proposed in this paper are poorly described and defined. What do the superscripts "size 1" and "size 2" mean? The BOI indices may be of some utility but again, like all similar indices based on empirical coefficients of total SPM, they are strictly qualitative in nature. The unknowns in the bulk coefficients in their definition will always cast doubt on their interpretation if ancillary evidence on PSD and composition are not available.

BOI indexes are not anymore part of the manuscript

Page 11, lines 20-21. Suggested wording: These relationships will be useful in investigating local and regionally-limited relationships and properties of SPM. Without separate independent studies of true optical properties of PIM and POM, and of PSD, these relationships will remain problematical.

We added the first sentence to the end of the conclusions paragraph

Technical Corrections: Page 2, line 5. Bowers et al. (2009) reported estimates of mass-specific scattering coefficients of and biogeo-physical characteristics of PIM, not SPM.

Ok corrected Page 6, line 26. Replace "properties" with "coefficients." done

Page 9, lines 9-10. The English usage here is nearly incomprehensible. Correct this and similar constructions with a native speaker of English.

Done

Pages 19-30. The tables presented here are nearly impossible to interpret. The table extensions to multiple pages have incomplete columns and no captions to the columns. The the table captions are limited and cryptic.

Many tables were removed to simplify content. Also, more labels were added to identify columns

References: Bohren, C.H. and D.R. Huffman. 1983. Absorption and Scattering of Light by Small Particles. John Wiley and sons, New York. Xiv + 530 pp. Chavez, F.P., Buck, K.R., Bidigare, R.R., Karl, D.M., Hebel, D., Latasa, M., and Campbell, L. 1995. On the chlorophyll a retention properties of glass-fiber GF/F filters. Limnol. Oceanogr., 40: 428–433. Johnson, B. D. and P. J. Wangersky. 1985. Seawater filtration: Particle flow and impaction considerations. Limnol. Oceanogr. 30: 966-971, doi:10.4319/lo.1985.30.5.0966. Richter, S.J. and R.H. Stavn. 2014. Determining functional relations in multivariate oceanographic systems: Model II multiple linear regression. Journal of Atmospheric and Oceanic Technology, 31: 1663-1672. Risović, D. 2002. Effect of suspended particulate-size distribution on the backscattering ratio in the remote sensing of seawater. Appl. Opt., 41(33): 7092-7101. Sheldon, R.W. 1972. Size separation of marine seston by membrane and and glass-fiber filters. Limnol. Oceanogr.,17(6): 494-498. Sheldon, R.W. and W.H. Sutcliffe, Jr. 1969. Retention of marine particles by screens and filters. Limnol. Oceanogr., 14(3): 441-444. Zhang, X., Stavn, R.H., Falster, A.U., Rick, J.J., Gray, D. and R.W. Gould, Jr. 2017. Size distributions of coastal ocean suspended particulate inorganic matter:Amorphous silica and clay minerals and their dynamics. Estuarine, Coastal and Shelf Science. DOI: 10.1016/j.ecss.2017.03.025

---

## Editor Decision (ED1)

[revised manuscript text omitted]

fig. 1

[Figure]

Fig. 2

[Figure]

5    Fig. 3

[Figure]

5    Fig. 4

[Figure]

fig. 5

[Figure]

Fig. 6

---

## Author Response (AR2)

**Reviewer #1**

Main comments

The manuscript has been re-organized and significantly improved since the first version. Some valuable work analyses are presented and discussed, based on shipborne measurements made on water samples collected in complex estuarine environments. At least part of the results obtained are (potentially) interesting for a better understanding of relationships between the optical, physical and biogeochemical properties of particles in suspension in natural waters. The discussion at the end of the study is rather speculative, notably due t the lack of complementary measurements (organic carbon, phytoplankton pigments, iron) and/or theoretical computations.
I still have many detailed comments concerning:
1. the section 'Data and methods' in which (i) the different parameters must be better defined, with appropriate physical units, (ii) the processing of optical and biogeo-chemical measurements must be clarified (see detailed comments below)
2. the section 'Results' in which could be partly re-organized for better logics
3. the section 'Discussion' which too speculative and could be shortened.

**We addressed the concerns of points 1-3. Also, the analysis of mass-normalized optical coefficients for PIM and POM was removed given the relatively small number of samples and large error of estimates.**

Therefore, I recommend a major revision of the manuscript.

Detailed comments

Page 5 lines 20-25
The absorption-beam attenuation meter (ac-s, WetLabs) is already calibrated by Wetlabs in pure water, so that it measures light absorption and attenuation on top of pure water contributions. Therefore, you do not have to subtract pure water absorption and attenuation coefficients from your measurements. Please correct in the text.
**we clarified that pure seawater contributions are removed during the calibration of the ac-s instrument**

Also why not applying the "proportional" method recommended by Wetlabs, that developed by: J. R. V. Zaneveld, J. C. Kitchen, and C. C. Moore, "Scattering error correction of reflecting tube absorption meters," Proc. SPIE 2258, 44-55 (1994).

**this point was explained in previous corrections. See below**
**Zaneveld et al. (1994) proposed the residual scattering correction method proportional to wavelength. This is analogous to the flat baseline correction but it assumes a spectral dependency on scattering errors. This approximation is in debate. Some studies have reported a spectral dependency on volume scattering functions or particulate backscattering ratios (Chami et al., 2006; McKnee et al.,**

**2009;McKnee et al., 2013). But there is doubt regarding if this assumption can be generalized (McKnee et al., 2013).**

Page 6, line 16
You should introduce/define here the Junge exponent often used to describe the size distribution of marine particles

**it was defined before since it needed to be discussed in introduction**

Why 35.17um?

**because it is the midpoint of the logarithmic size range**

Page 6, Line 27
'the spectral slope of the particulate attenuation coefficient is negatively related with the mean particle size for particles smaller than 20 mm.'
What is the reference for that??

**sorry but we couldn't find this sentence along the text**

Page 7, lines 4-6
Untrue: if SPM includes a significant contribution of phytoplankton (chlorophyll-a) to light absorption, which kas absorption peaks at440 nm and 675 nm.
Your Equation (5) only applies in the case of non-algal particles.
Please correct this.
**done**

Equations (6-7): units?
**this is explicit in the list of acronyms, however we added units too before describing the equations**

Lines 10-15: totally unclear whereas you define the mass-specific absorption and scattering coefficients (in m2 /g) of the mass-specific absorption cross sections for mineral and organic fractions of SPM??? And what is the difference between these coefficients. Please clarify these definitions and always provide the appropriate physical units.

**the definition of these sigmas or cross sections are in section 2.7**
**the units again are in table 1 but we placed units too along the text**

where m is the mass in g m-3 for each size class i….? g m-3 is a mass per unit of volume, which means a concentration of SPM…… not surprising I do not understand the definitions of your parameters!!

**sorry, good catch. now missing text added**

>> Please rewrite sections 2.5, 2.6 and 2.7 and clarify the definitions of the parameters you compute then use to describe how the SPM size and chemical composition influence the SPM optical properties. As it I found it very confusing.
**these sections were clarified in terms of units and re-organized**

Page 8, section 3.1
Please use either decimal numbers (e.g., 0.0111) or percentages (11%) to report the contributions f small/large and organic-rich/poor particles on the measured optical properties. This should make the text easier to read.

**done**

Line 10
'The uncertainty of x calculations, as estimated from 2 standard errors, varied between 1.6 and 10.2%'
>> To be computed and reported in the section 'Data and Methods'

**done**

page 8
'3.1 Spatial variability of microphysical properties of SPM'
What do you mean exactly by 'microphysical'?

**microphysical properties is a term also used in atmospheric optics for aerosols and it refers to physical properties at the microscopic level (e.g., size, shape).**
**The opposite would be macroscopic level or macrophysical properties (e.g., clouds in in atmospheric sciences). In oceanography, an aggregate should be considered as a macrophysical phenomenon.**

Overall I do not clearly understand the logics with the other two Results sections entitled:
'3.2 Mass-specific optical properties of SPM' and '3.4 Optical proxies' (should be 3.3)
I would rather suggest to
1) report the mass-specific coefficients and corresponding spectral slopes, analyse their spatial variations and compare them to values already published in similar environments,

**we appreciate your suggestion but the way we organized the data makes sense for us. Mass-normalized optical properties and optical proxies are two different topics in the publication that are clearly separated.**
**We can not perform spatial analysis of mass-normalized optical properties with respect to other pubs because we dont have really have resolution for chemical fractions (only 2 regions) and for size fractions there are NO studies in the literature other than ours for comparing. We compared geographic variations on mass-normalized optical coefficients in Table 4**

2) analyse the influence of SPM size on the mass-specific absorption and scattering coefficients, then

**the size and particle composition effects are discussed when talking about the mass-normalized optical properties for different size fractions of SPM in Table 2. We cant with chemical fractions**
**the focus of this study is examine the influence of PSD and particle composition on optical properties of SPM fractions and not total SPM.**

3) analyse the influence of SPM composition (PIM, POM) on the mass-specific absorption and scattering coefficients.

**the focus of this study is examine the influence of PSD and particle composition on optical properties of SPM fractions and not total SPM.**

page 8, section 3.2
Are your mass-specific optical properties of SPM in good agreement with values reported in similar environments? Please discuss this first.

**this is discussed in section 4.3. In general there is an agreement with published values reported in other environments for aspm\* and bspm\* as derived from GF/F filtrations. However, differences were detected for optical cross sections of chemical fractions. This is also discussed.**

Why finally presenting results for the 400-650 nm spectral interval?

**the spectral range of ac-s measurements in this study is from 400.3 to 747.5 nm. So we don't have wavelengths below 400 nm in the UV spectral range.  In the NIR, only wavelengths below 710 nm were examined due to the large error of ac-s determinations at longer wavelengths.**

Line 23
Here I do not understand why you report values in 1/m. Based on the definition of this parameter (Equation 6, i.e. absorption coefficient (in 1/m) divided by a concentration of SPM (in g m-3) we would expect values in m2 g-1….??? Please check and clarify all

**this.sorry about the typo. now units are corrected**

Line 25
'Similar to ai\*, highest bi\* values (up to 5.7 m2 g-1 for l'
and here you report values in m2 g-1…..

**these are mass-specific optical coefficients and the units are correct**

Line 26

'corresponded with size fractions having particles with the smallest and the largest diameter'
What do you mean?? Please rewrite

**these coefficients were measured in size-fractionated samples corresponding to particle size ranges of 0.2-0.7 microns and >10 microns.**

Lines 27-28
'In general, the spectral slope of $bi^*$ was very variable in all size fractions (-6 10-5 to 6.28 10-3 nm-1) with the greatest spectral changes associated to particulates greater than 10 mm.'
??? Why suddenly talking about the spectral slope of the SPM scattering coefficient? This spectral slope was not even defined!

that's true. this result is disconnected and was part of an old manuscript version. it is irrelevant and was deleted

Page 9, line 22
'Also for SPM fraction having the largest particulates'
?? largest grain size?
**we rewrote the senetence: 'Also for the size fraction of SPM with a grain size >10 microns'**
* * *
**Reviewer #2**

General comments

Page 2, lines 1-2 of original manuscript. Algorithms based only on CSPM will never have the accuracy required for optical inversions because SPM is undefined optically, an unknown mixture of inorganic and organic matter. Therefore partition of SPM into at least major chemical composition classes (PIM and POM) and estimation of size distribution are required independently for optically-based remote sensing algorithmsof primary productivity and suspended mineral dynamics of "disappearing shorelines" etc.

**we added this important point in the introduction**

Estimating SPM from summing PIM and POM estimates has at least a definable and reproducible measurement error. Basically there is a confusion here between systematic errors and measurement errors. The ubiquitous (and wrongly proposed in my opinion) algorithms based on just SPM will provide precisely calculated inaccurate estimates.

**also, we emphasized this point in introduction**

Author's response for Page 3, lines 15-20 of original manuscript:

No, different filters were used in this study to fractionate 0.4-0.7 and 0.7-10 micron fractions This sentence was clarified as:

Size fractionation of SPM into four size classes (>10 μm, 0.7-10 μm, 0.4-0.7 μm, and 0.2-0.4 μm) was done after sequentially filtering the original samples through preweighted membranes having a diameter of 47 mm and a pore size of 10 μm (Whatman, polycarbonate), 0.7 μm (GF/F, Whatman, glass fiber), 0.4 μm (Whatman, polycarbonate), and 0.2 μm (Nucleopore, polycarbonate), respectively. Also, we add the whatman GF/F was used for PIM and POM determinations.

What is unclear here is why the glass fiber GF/F filters were used for both "size fractionation" and retrieval of "total SPM." If the GF/F filters are removing all the SPM then sequential fractionation should not yield anything removed by filters smaller than the 70 micron nominal pore size of the GF/F filters.

**this paragraph was clarified. GF/F filters were only used for SPM fractions. Total SPM was computed by adding the weight of the 4 size fractions. Thus, total SPM represent particulates > 0.2 microns**

Page 7, line 19 of revised manuscript. The authors should quote Richter and Stavn (2014) which actually gives instruction on accomplishing Model II multiple regression utilizing R code
**done**

Page 13, lines 12 and 19, Page 14, lines 29 and 30 of revised manuscript. Don't the p values quoted in these lines indicate that the correlations quoted are not significant?

**there was a typo here and the sentence was rewritten**

Page 11, lines 20-21 of original manuscript. Suggested wording: These relationships will be useful in investigating local and regionally-limited relationships and properties of SPM. Without separate independent studies of true optical properties of PIM and POM, and of PSD, these relationships will remain problematical.
We added the first sentence to the end of the conclusions paragraph
I recommend both sentences at the end of the conclusions.
**done**

---

## Author Response (AR3)

October, 9, 2017

**Dear Dr. Emmanuel Boss**

**Editor Associate**

**Biogeosciences**

The whole manuscript was revisited and corrected according to your last suggestions.

The text was shortened and clarified and the speculations about iron or other topics were eliminated. Figure 6 was eliminated because it was redundant. Lastly, the english style was carefully reviewed and corrected as needed.

We are very grateful for your help throughout the reviewing process

We hope this version meets the quality of your journal and we look forward to receive a favorable response

My best regards,

Dr. Martin Montes

Professor

ISMER

University of Quebec at Rimouski

[revised manuscript text omitted]

fig. 1

[Figure]

Field Code Changed

[Figure]

Fig. 2

[Figure]

Field Code Changed

[Figure]

Fig. 3

[Figure]

[Figure]

Field Code Changed

Fig. 4

[Figure]

fig. 5

[Figure]

Fig. 6

---

## Editor Decision (ED3)

[revised manuscript text omitted]

fig. 1

[Figure]

Fig. 2

[Figure]

Fig. 3

[Figure]

Fig. 4

---

## Author Response (AR4)

October, 19, 2017

**Dear Dr. Emmanuel Boss**

**Editor Associate**

**Biogeosciences**

The english style of the manuscript was reviewed by a native english speaker. Figure 6 was removed and the text was shortened. Speculations were removed.

Thank you again for your support and advice during the entire review.

My best regards,

Dr. Martin Montes

Professor

ISMER

University of Quebec at Rimouski

[revised manuscript text omitted]

---

## Author Response (AR5)

October, 25, 2017

**Dear Dr. Emmanuel Boss**

**Editor Associate**

**Biogeosciences**

Please receive the final corrected version the manuscript 'Optical properties of size fractions of suspended particulate matter in littoral waters of Quebec' to be published in Biogeosciences. Below the last modifications following your suggestions.

1. Regarding the use of PSD slope as an index of particle size. The index D50 was not used in this manuscript but it was found to be correlated with LISST-derived $\xi$ when processing data for another publication (Mohammadpour et al., 2015)

Mohammadpour G., Montes-Hugo M.A., Stavn R., Gagne J.P., Larouche L. 2015. Particle composition effects on MERIS-derived SPM: a case study in the Saint Lawrence Estuary. Canadian Journal of Remote Sensing, 41, 514-524.

2. LISST-derived $\xi$ is not different between regions but when we compare the variability between samples (not regions) then the variability of PIM/SPM is smaller than that attributed to LISST-derived $\xi$.

3. Very high $bi^*$ >10 µm values were corrected due to major errors on weight of size fractions of SPM

4. The spectral slope of $c_{SPM}$ ($\gamma$) can not be compared between PSICAM and ac-s measurements because PSICAM only measures absorption values not beam attenuation values.

5. The errors on mass-specific coefficients is even larger (>15%) than those corresponding to the optical coefficients due to the additional uncertainty associated to the weight of particulates in $a_{SPM}^*$ and $b_{SPM}^*$ determinations

Thank you again for your support and advice during the entire review.

My best regards,

Dr. Martin Montes

Professor

ISMER

University of Quebec at Rimouski